# A structural mechanism for phosphorylation-dependent inactivation of the AP2 complex

Edward A Partlow[1†], Richard W Baker[2†*], Gwendolyn M Beacham[1],
Joshua S Chappie[1], Andres E Leschziner[2,3*], Gunther Hollopeter[1*]

[1]Department of Molecular Medicine, Cornell University, New York, United States;
[2]Department of Cellular and Molecular Medicine, School of Medicine, University of California, San Diego, La Jolla, United States; [3]Section of Molecular Biology, Division of Biological Sciences, University of California, San Diego, La Jolla, United States

*For correspondence:
ribaker@ucsd.edu (RWB);
aleschziner@ucsd.edu (AEL);
gh383@cornell.edu (GH)

†These authors contributed equally to this work

Competing interests: The authors declare that no competing interests exist.

**Abstract** Endocytosis of transmembrane proteins is orchestrated by the AP2 clathrin adaptor complex. AP2 dwells in a closed, inactive state in the cytosol, but adopts an open, active conformation on the plasma membrane. Membrane-activated complexes are also phosphorylated, but the significance of this mark is debated. We recently proposed that NECAP negatively regulates AP2 by binding open and phosphorylated complexes (Beacham et al., 2018). Here, we report high-resolution cryo-EM structures of NECAP bound to phosphorylated AP2. The site of AP2 phosphorylation is directly coordinated by residues of the NECAP PHear domain that are predicted from genetic screens in *C. elegans*. Using membrane mimetics to generate conformationally open AP2, we find that a second domain of NECAP binds these complexes and cryo-EM reveals both domains of NECAP engaging closed, inactive AP2. Assays in vitro and in vivo confirm these domains cooperate to inactivate AP2. We propose that phosphorylation marks adaptors for inactivation.

DOI: https://doi.org/10.7554/eLife.50003.001

## Introduction

Clathrin-Mediated Endocytosis (CME) enables cells to dynamically regulate the composition of the plasma membrane and mediate uptake of transmembrane cargo, such as ligand-bound receptors. This process is orchestrated by the clathrin Adaptor Protein two complex (AP2), which interacts with much of the endocytic machinery and functions during the earliest stages of CME (*Mettlen et al., 2018*). Inactive, closed AP2 is initially recruited to the cytosolic face of the plasma membrane through its interaction with PhosphatidylInositol-4,5-bisPhosphate ($PIP_2$) (*Collins et al., 2002*; *Höning et al., 2005*). At the plasma membrane, conformational rearrangement of AP2 to an active, open conformation is promoted by membrane-associated muniscin proteins (*Henne et al., 2010*; *Hollopeter et al., 2014*; *Umasankar et al., 2014*) (*Figure 1A*). Conversion of AP2 to the open conformation exposes a second binding site for $PIP_2$ that stabilizes membrane engagement (*Kadlecova et al., 2017*), and also reveals binding sites for clathrin and membrane-embedded cargo (*Jackson et al., 2010*; *Kelly et al., 2014*). These interactions allow AP2 to function as the central regulatory hub of clathrin-coated vesicle formation (*Kirchhausen et al., 2014*).

After opening, AP2 is phosphorylated on the mu subunit (*Conner et al., 2003*; *Jackson et al., 2003*; *Pauloin et al., 1982*), but the role of this mark remains poorly defined. Some data suggest that phosphorylation of AP2 enhances binding to $PIP_2$ and cargo (*Fingerhut et al., 2001*; *Höning et al., 2005*; *Ricotta et al., 2002*). Additionally, mutation of the phosphorylated threonine

(mu T156A) or addition of kinase inhibitors have been shown to inhibit transferrin uptake in tissue culture (*Olusanya et al., 2001*), implying that phosphorylation promotes AP2 activity. Two kinases have been shown to phosphorylate AP2 (mu T156) in vitro: AP2-Associated Kinase (AAK1) (*Conner and Schmid, 2002*) and cyclin-G associated kinase (GAK) (*Umeda et al., 2000*). Curiously, studies involving these kinases suggest that phosphorylation may function in an inactivation pathway, as AAK1 appears to inhibit endocytosis using in vitro assays (*Conner and Schmid, 2002*), and GAK seems to function in vesicle uncoating (*Taylor et al., 2011*). Whether phosphorylation is an activating or inactivating mark, and whether it functions in multiple stages in the endocytic pathway remains to be determined.

While the pathways of clathrin adaptor activation have been well characterized both structurally and biochemically (*Collins et al., 2002*; *Kelly et al., 2008*; *Kelly et al., 2014*; *Jackson et al., 2010*; *Ren et al., 2013*; *Jia et al., 2014*), it is unknown whether inactivation of AP2 is also a regulated process. Adaptor inactivation likely occurs throughout the endocytic cycle. High-resolution imaging reveals that many endocytic pits abort prematurely, presumably when a requirement for activation is unmet, such as an absence of $PIP_2$, cargo, or muniscin (*Cocucci et al., 2012*; *Kadlecova et al., 2017*). Abortive events could represent a mechanism to limit futile vesicle formation in the absence of cargo or prevent ectopic budding from off-target membranes lacking $PIP_2$. Additionally, when adaptors are removed from vesicles prior to fusion with target organelles, they must also revert to the cytosolic, inactive state. Adaptor uncoating appears to require $PIP_2$ phosphatase activity (*Cremona et al., 1999*; *He et al., 2017*), but it is not known whether $PIP_2$ hydrolysis is sufficient to uncoat AP2, as cytosol and ATP are required in vitro (*Hannan et al., 1998*). It remains to be determined whether adaptor inactivation occurs via stochastic disassembly of endocytic pits (*Ehrlich et al., 2004*), or if it is driven by regulated mechanisms, such as an endocytic checkpoint to ensure cargo incorporation (*Loerke et al., 2009*; *Puthenveedu and von Zastrow, 2006*).

We recently found that NECAP appears to act as a negative regulator of AP2 in vivo (*Beacham et al., 2018*). NECAP is a coat-associated protein that binds to the alpha appendage and beta linker regions of AP2 to facilitate endocytic accessory protein recruitment to sites of endocytosis (*Ritter et al., 2003*; *Ritter et al., 2004*; *Ritter et al., 2007*; *Ritter et al., 2013*). In *C. elegans*, loss of the muniscin, *fcho-1*, causes AP2 to dwell in an inactive, closed state (*Hollopeter et al., 2014*). Deletion of the gene encoding NECAP (*ncap-1*) in *fcho-1* mutants restores AP2 to an active, open and phosphorylated state, suggesting that NECAP counterbalances AP2 activation. Consistent with this model, NECAP binds open and phosphorylated AP2 complexes in vitro, however it is unclear how NECAP recognizes the open and phosphorylated states and whether NECAP can directly close these complexes.

Here, we determined cryo-EM structures of NECAP-AP2 complexes which show that NECAP clamps AP2 complexes into the closed, inactive conformation. This mechanism requires coincident binding of two domains of NECAP, one of which confers specificity for open complexes, and another that detects AP2 phosphorylation. Our structures are supported by in vitro biochemistry along with functional assays and unbiased genetic screens in *C. elegans*. Importantly, the site of AP2 phosphorylation is directly bound by NECAP, defining phosphorylation as a key step in the dynamic regulation of AP2 inactivation.

## Results

### Structural basis for recognition of phosphorylated AP2 by NECAP

We previously demonstrated that NECAP binds to the phosphorylated AP2 core (*Beacham et al., 2018*). NECAP is a ~ 29 kDa, soluble protein composed of three domains: an N-terminal Pleckstrin Homology with ear-like function ($NECAP_{PHear}$), a central Extended region of conservation ($NECAP_{Ex}$), and a C-terminal domain with low conservation ($NECAP_{Tail}$) (*Figure 1B*) (*Ritter et al., 2007*; *Ritter et al., 2013*). While there are two paralogues of NECAP in vertebrates, we have previously shown that they function equivalently to bind phosphorylated and open AP2 complexes in vitro and rescue loss of NECAP in *C. elegans* (*Beacham et al., 2018*). In this work, we use human or mouse NECAP2 for all experiments because of their ease of purification and stability. To narrow down

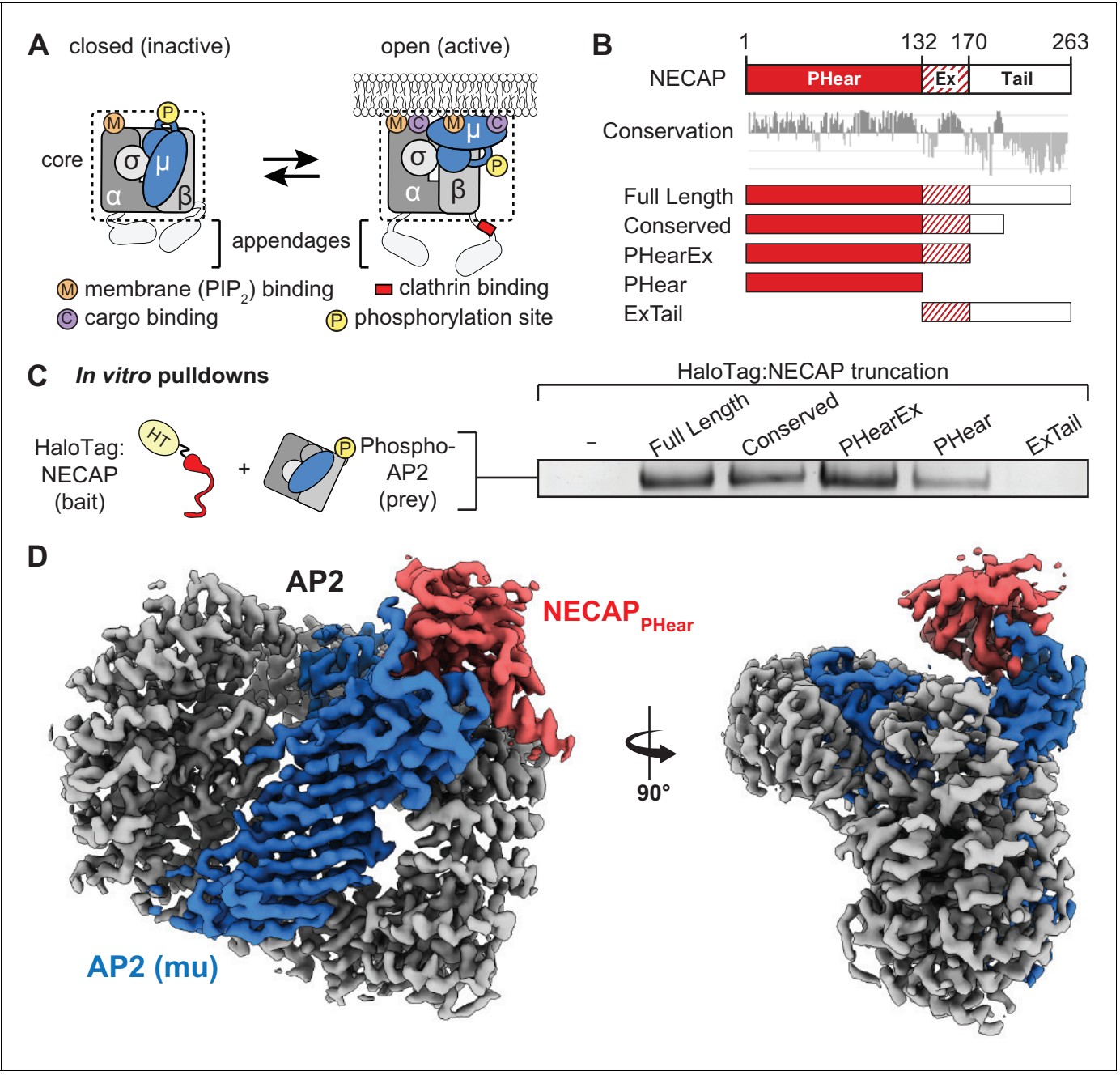

**Figure 1.** NECAP PHear domain binds phosphorylated AP2 core. (A) AP2 comprises four subunits: alpha (α), beta (β), mu (μ), and sigma (σ). The complex also comprises a structured 'core' (dashed box) connected by flexible linkers to appendages on the alpha and beta subunits. Binding sites for AP2 substrates are indicated. PIP$_2$, PhosphatidylInositol-4,5-bisPhosphate. (B) (Top) NECAP domain organization (numbers are for human NECAP2). (Center) Conservation scores of NECAP residues calculated using the ConSurf server (*Ashkenazy et al., 2010*; *Celniker et al., 2013*). (Bottom) Truncation constructs used in this work. (C) Binding analysis of NECAP truncations compared to HaloTag control (-). HT, HaloTag; phosphoAP2, phosphorylated AP2 core. Representative image of three technical replicates. (D) Cryo-EM map of the phosphoAP2-NECAP complex (PDB 6OWO). Red: NECAP; Blue: AP2 mu; Gray: AP2 alpha, beta, sigma. See also *Figure 1—figure supplement 1* and *Figure 1—figure supplement 2*.
DOI: https://doi.org/10.7554/eLife.50003.002

The following figure supplements are available for figure 1:

**Figure supplement 1.** Classification, signal subtraction, and refinement of phosphoAP2 bound to NECAP.
DOI: https://doi.org/10.7554/eLife.50003.003

**Figure supplement 2.** Comparison of NECAP solution structure and phosphoAP2-NECAP cryo-EM structure.
DOI: https://doi.org/10.7554/eLife.50003.004

*Figure 1 continued on next page*

*Figure 1 continued*

**Figure supplement 3.** Model building for mu pT156 linker region.
DOI: https://doi.org/10.7554/eLife.50003.005

which domain of NECAP binds phosphorylated AP2, we performed in vitro pulldown assays using NECAP truncations and phosphorylated AP2 cores lacking appendages (phosphoAP2; rodent; boxed in *Figure 1A*). Our analysis showed that NECAP$_{PHear}$ is necessary and sufficient to bind phosphorylated AP2 in vitro (*Figure 1C*).

Phosphorylation is thought to stabilize the open conformation of AP2 and we previously showed that NECAP can bind open AP2 that is not phosphorylated. To understand whether NECAP binds to the site of phosphorylation or a conformation induced by phosphorylation, we determined a ~ 3.2 Å cryo-EM structure of the phosphorylated AP2 core bound to full-length NECAP2 (mouse; *Figure 1D*, *Figure 1—figure supplement 1*, *Figure 1—figure supplement 2*, *Table 1*). Globally, AP2 has adopted a conformation similar to the crystal structure of the closed complex (PDB 2VGL; *Figure 1—figure supplement 2A*) (*Collins et al., 2002*), which is inactive due to the occlusion of binding sites for cargo and the plasma membrane. We observe density for the entire AP2 core, with additional density contacting the mu subunit (*Figure 1—figure supplement 2A*). Most of this additional density can be attributed to NECAP$_{PHear}$, due to similarity with a solution NMR structure of the mouse NECAP1 PHear domain (PDB 1TQZ; *Figure 1—figure supplement 2B,C*) (*Ritter et al., 2007*). The remaining unassigned density at this interface extends from the amino terminus of residue 159 of mu, and can be attributed to the phosphorylated mu linker (amino acids 154–158, *Figure 1—figure supplement 2C*). The register leading up to this new density is based on a previous crystal structure (PDB 2VGL), and is confirmed by the positions of several landmark residues near T156 that fit the density well. A detailed schematic for building the critical T156 residue in our model is shown in *Figure 1—figure supplement 3*. Our structure is of sufficient quality to build a near-complete molecular model for the phosphoAP2-NECAP complex, which shows that residues in NECAP$_{PHear}$ interact directly with the phosphorylated mu T156 (pT156; see Figure 3B below). These data show that the phosphorylated complex can exist in a closed conformation and suggest that the mechanism of NECAP binding to open complexes that are not phosphorylated must be fundamentally different.

## A genetic screen in *C. elegans* identifies mutations that disrupt the NECAP-AP2 interface

Our structural data using vertebrate proteins imply that NECAP$_{PHear}$ and its interaction with mu pT156 is central to the function of NECAP. To test this hypothesis, we turned to *C. elegans*, where we devised a genetic strategy to specifically isolate critical residues required for NECAP interaction with AP2, while avoiding mutations that destabilize the NECAP protein (*Figure 2A*). This screen was based on our previous screen for mutations that suppressed the morphological and fitness defects of *fcho-1* mutants and restored the active, open state of AP2. In the original screen, the predominant mutations identified were either gain-of-function mutations in AP2 subunits that destabilize the closed conformation (*Hollopeter et al., 2014*) or null mutations in the gene encoding NECAP, *ncap-1* (*Beacham et al., 2018*). To identify rare mutations that specifically disrupt the functional interface between AP2 and NECAP, we repeated the screen with a fluorescent tag on NECAP to enable secondary classification of suppressed animals based on fluorescent hallmarks (*Figure 2A*). Because gain of function mutations in AP2 result in NECAP recruitment to the nerve ring, a membranous tissue with a high concentration of AP2 (*Beacham et al., 2018*), we visually eliminated suppressed animals that had fluorescent nerve rings. We also eliminated suppressed animals that were no longer fluorescent, as these likely harbored null mutations in NECAP. We reasoned that the remaining animals, which suppressed *fcho-1* phenotypes without altering the fluorescent signal, might possess missense mutations that prevented NECAP from binding AP2. Worms that met these requirements were selected, and the genes encoding NECAP and AP2 were sequenced. We isolated mutations in NECAP$_{PHear}$ and AP2 mu that were previously proposed to disrupt NECAP-AP2 interaction, as well as new potential interface mutations in the AP2 alpha and beta subunits (*Figure 2B and C*). Strikingly, the overwhelming majority of mutations (8 out of 13) isolated were in mu T156, suggesting that phosphorylation of this residue generates the substrate for NECAP activity.

**Table 1.** Cryo-EM data collection, refinement, and validation statistics.

| | pAP2-NECAP 'unclamped' PDB ID: 6OWO EMDB ID: EMD-20215 | pAP2-NECAP 'clamped' PDB ID: 6OXL EMDB ID: EMD-20220 |
|---|---|---|
| **Data Collection** | | |
| Microscope | Talos Arctica | Talos Arctica |
| Camera | K2 summit | K2 summit |
| Camera Mode | Super-Resolution | Counting |
| Magnification | 36,000 | 36,000 |
| Voltage (kV) | 200 | 200 |
| Total electron exposure (e-/Å2) | 60 | 50 |
| Exposure rate (e-/pixel/sec) | 6.67 | 6.43 |
| Defocus Range (um) | 0.6–2.5 | 0.6–2.5 |
| Pixel Size (Å/pixel) | 0.58 | 1.16 |
| Micrographs collected (no.) | 1092 | 1497 |
| Micrographs used (no.) | 944 | 1126 |
| **Reconstruction** | | |
| 3D Processing Package | Relion 3 | cryoSPARC v2 |
| Total Extracted picks (no.) | 890,658 | 717,231 |
| Refined particles (no.) | 490,560 | 388,962 |
| Final Particles (no.) | 71,571 | 324,922 |
| Symmetry | C1 | C1 |
| Resolution (global) (Å) | 3.2 | 3.5 |
| FSC 0.143 (unmasked/masked) | 4.1/3.2 | (4.2/3.5) |
| Local resolution range (Å) | 3.1–4.2 | 3.2–6.1 |
| Map sharpening $B$-factor | −29 | −160 |
| **Refinement** | | |
| Model refinement package | Rosetta, phenix.real_space_refine | Rosetta, phenix.real_space_refine |
| Model composition | | |
| Nonhydrogen atoms | 14,280 | 14,418 |
| Protein residues | 1785 | 1814 |
| $B$ factors (Å2) | | |
| Protein residues | 62.53 | 62.72 |
| R.m.s. deviations | | |
| Bond Lengths (Å) | 0.009 | 0.01 |
| Bond angles (°) | 1.182 | 1.253 |
| **Validation** | | |
| MolProbity (score/percentile) | 1.4/97[th] | 1.75/87[th] |
| Clashscore (score/percentile) | 3.92/96[th] | 6.62/88[th] |
| Poor rotamers (%) | 0.25 | 0.06 |
| CaBLAM outliers (%) | 1.34 | 1.38 |
| Ramachandran plot | | |
| Favored (%) | 96.57 | 94.36 |
| Allowed (%) | 3.32 | 5.64 |
| Disallowed (%) | 0.11 | 0 |
| EMRinger score | 3.33 | 3.05 |
| Map CC (*CCmask*) | 0.857 | 0.815 |

DOI: https://doi.org/10.7554/eLife.50003.006

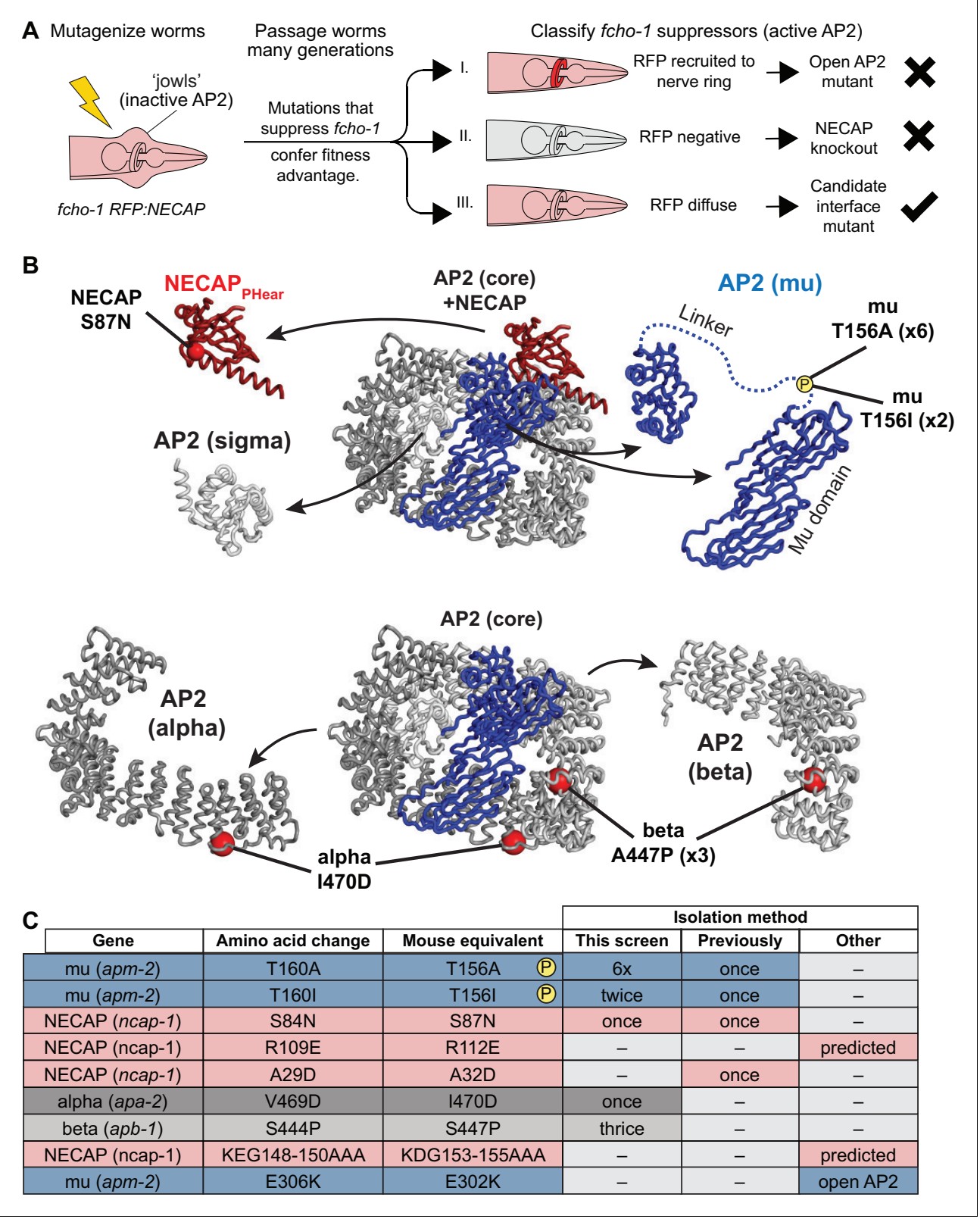

**Figure 2.** Genetic screen for NECAP-AP2 interface mutations. (**A**) Schematic of genetic screen to identify residues important for AP2-NECAP binding. FCHo mutants (*fcho-1*) exhibit growth defect and 'jowls' phenotype due to inactive AP2 (cartoon is anterior of worm). Fluorescent tag on NECAP (RFP: NECAP) enables visual categorization of suppressor mutations that restore AP2 activity. (**B**) Mutations identified in (**A**) mapped as red spheres onto the

*Figure 2 continued on next page*

*Figure 2 continued*

subunits of the closed AP2 crystal structure (2VGL) and our NECAP_PHear cryo-EM structure (PDB 6OWO). (C) Table of *C. elegans* mutations and their vertebrate equivalents referenced in this manuscript.

DOI: https://doi.org/10.7554/eLife.50003.007

## Coordination of phosphorylated AP2 by NECAP is required for inactivation

To understand how the mutated residues isolated in our screens disrupt NECAP function, we mapped the vertebrate equivalents onto our cryo-EM structure (*Figure 3A,B*). Mutations of T156 itself disable phosphorylation completely, and the two residues in NECAP_PHear (A32 and S87) lie within 10 Å of mu T156, which suggests that they disrupt coordination of the phosphate group and explains how they might break the AP2-NECAP interaction in vivo (throughout the text, all residue numbers are for vertebrate proteins, see *Figure 2C* for *C. elegans* equivalents). We also noticed that NECAP R112 forms electrostatic interactions with the T156 phosphorylation mark and may be required for binding (*Figure 3B*). To test the hypothesis that coordination of mu T156 phosphorylation is necessary for NECAP function, we introduced these mutations into recombinant vertebrate complexes and *C. elegans* strains and tested each using a panel of in vitro and in vivo assays. A bona fide interface mutant should reduce the function of NECAP in multiple assays, but not necessarily to the same degree in every assay. This is because each assay exhibits a different linear range, and point mutations may affect *C. elegans* and vertebrate proteins differently.

As predicted, mutation of NECAP R112 disrupted binding in vitro, similar to either mutations in the mu linker (T156A) or NECAP_PHear (S87N) (*Figure 3C*). Consistent with the screen, the mutations suppressed the morphological and fitness defects of *fcho-1* mutants, as quantified by the number of days for a population to expand and consume a food source (*Figure 3D*). It is worth noting that mutation of R112 did not fully suppress the fitness defect, and thus would likely not have been isolated from our genetic screen. Additionally, chemical mutagenesis favors cytosine to thymine DNA transitions, thus disfavoring charge reversal of R112. Nonetheless, all three mutations reduced NECAP recruitment to the *C. elegans* nerve ring (*Figure 3E*) and resulted in accumulation of open AP2 complexes according to an in vivo protease sensitivity assay (*Figure 3F*). Together, our data show that NECAP_PHear binding to the phosphorylated threonine dictates AP2 inactivation.

While the AP2-NECAP_PHear interface is clearly important for inactivation, our cryo-EM structure does not explain why. NECAP_PHear only contacts the mu subunit of AP2, and there are no clashes when mu-NECAP_PHear is modeled as a rigid body into the crystal structure of open AP2 (PDB 2XA7; *Figure 3—figure supplement 1*). Additionally, NECAP_PHear does not block cargo- or PIP_2-binding sites and no obvious steric clashes would prevent AP2 from transitioning between open and closed states when bound to NECAP_PHear (*Figure 3—figure supplement 1*). One feature that might prevent NECAP_PHear binding to the open conformation is that mu T156 is packed against the beta subunit of AP2 in this structure (*Figure 3—figure supplement 1C*) (*Jackson et al., 2010*). This suggests that release and phosphorylation of the mu linker are important regulatory steps in NECAP recruitment. Nonetheless, while NECAP_PHear is required to inactivate AP2, binding of NECAP_PHear alone does not explain inactivation. Because mutations in NECAP_PHear do not disrupt binding to open, non-phosphorylated AP2 complexes (*Beacham et al., 2018*), we believe another domain of NECAP may contribute to its function.

## Membrane mimetics stimulate opening of AP2

To understand how NECAP recognizes open AP2 in the absence of phosphorylation, we needed to control and measure the conformation of AP2 in vitro. We used structural data to engineer a protease site on AP2 that is preferentially cleaved in the open state (*Aguilar et al., 1997*; *Hollopeter et al., 2014*; *Matsui and Kirchhausen, 1990*) (*Figure 4A*) and introduced a mutation in the mu subunit (mu E302K) that is known to promote the open conformation (*Hollopeter et al., 2014*). Despite this mutation, our AP2 complexes remained largely protease insensitive (i.e. closed) in the absence of other factors (*Figure 4B*). To produce open AP2, we turned to the observation that binding of AP2 to cargo is dramatically stimulated by the addition of long chain heparin (*Jackson et al., 2010*). Long anionic polymers are hypothesized to mimic negatively-charged PIP_2-

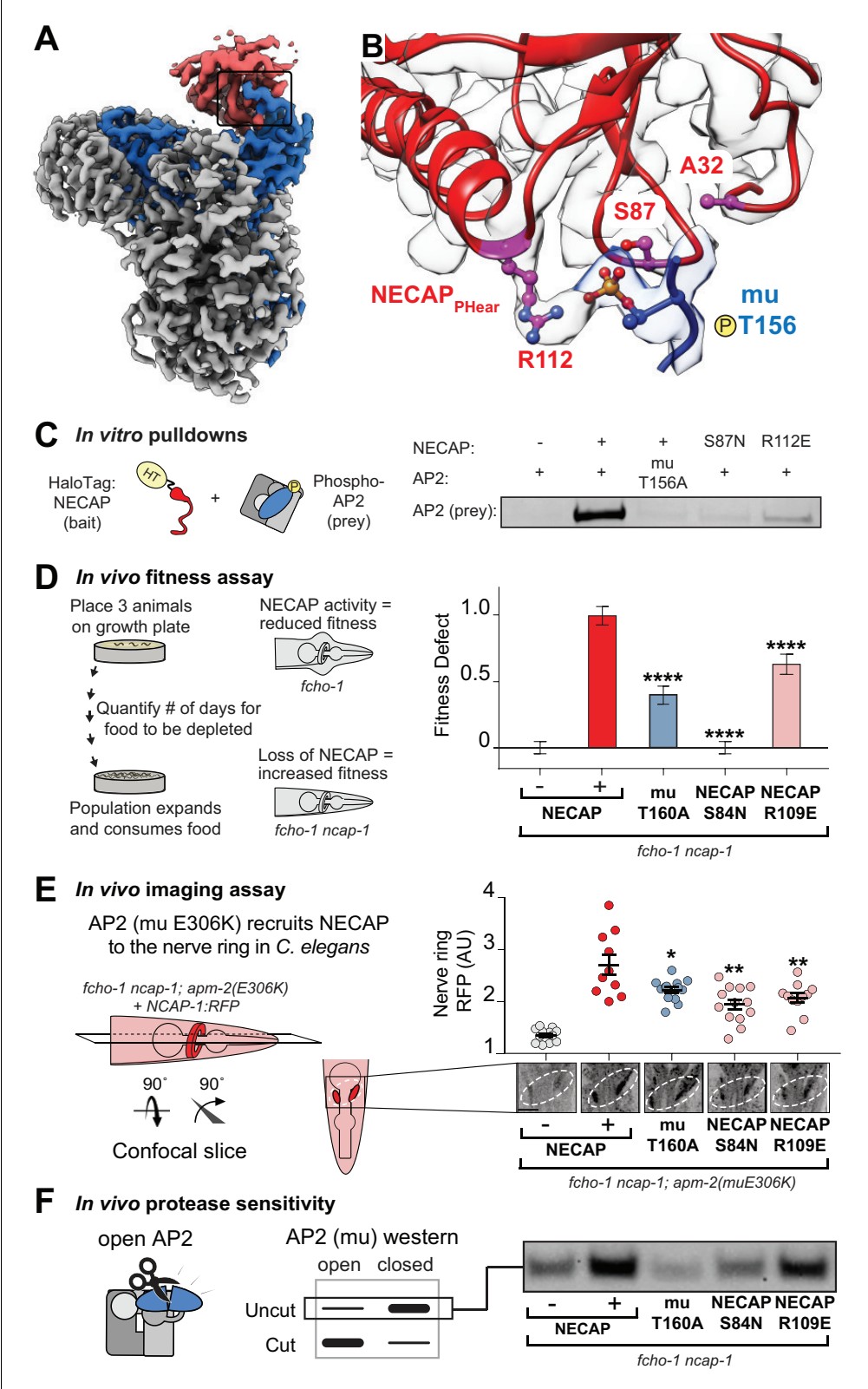

**Figure 3.** Coordination of phosphorylated AP2 by NECAP is required for inactivation. (**A**) Cryo-EM of phosphoAP2-NECAP complex, boxed region shown in (**B**). (**B**) phosphoAP2-NECAP interface as a ribbon diagram inside a transparent rendering of the cryo-EM map with ball-and-stick representation of relevant NECAP side chains and mu pT156. Red: NECAP; Blue: AP2-mu. See also *Figure 3—figure supplement 1*. (**C**) Binding analysis of interface mutants compared to HaloTag control (–). HT, HaloTag. Representative image of two technical replicates. (**D**) In the absence of
*Figure 3 continued on next page*

Figure 3 continued

NECAP (–), *fcho-1* mutants take about 4 days to proliferate and consume a bacterial food source (fitness defect = 0). Expression of NECAP (+) increases the number of days to about 8 (fitness defect = 1). Data for interface mutants were normalized to this fitness defect; n = 10 biological replicates. (E) (Left) In *fcho-1; apm-2 (E306K)* mutant worms, NECAP is recruited to the nerve ring. Interface mutants disrupt nerve ring recruitment, as quantified by in vivo confocal microscopy. (Right) Normalized RFP intensities plotted above representative confocal nerve ring images of ten biological replicates. (D–E) Error bars indicate mean ± SEM. Significance compared to NECAP (+); Student's t-test performed on raw data (D) or normalized data (E). *p<0.05, **p<0.01, ***p<0.001, ****p<0.0001. (F) In vivo protease sensitivity assay to probe AP2 conformation in genetic backgrounds indicated. In the absence of NECAP (–), AP2 is protease sensitive (open). Expression of wild type NECAP (+) results in protease resistant AP2 (closed). All strains lack *fcho-1*.
DOI: https://doi.org/10.7554/eLife.50003.008
The following figure supplement is available for figure 3:

**Figure supplement 1.** Comparison of NECAP binding site in open and closed AP2 conformations.
DOI: https://doi.org/10.7554/eLife.50003.009

containing membranes. To test whether these types of negatively charged macromolecules could affect AP2 conformation, we included them in our protease sensitivity assay. Indeed, two anionic polymers, heparin and nucleic acids, generated protease-sensitive complexes, suggesting that they stimulated a conformational rearrangement of AP2 to the open state (*Figure 4B*). Importantly, the $PIP_2$ mimetic, inositol hexakisphosphate (IP6), blocks the stimulatory effect of DNA on AP2, and is not sufficient to open the complex itself (*Figure 4B*), consistent with the model that open AP2 is stabilized by simultaneous engagement of multiple $PIP_2$ binding sites by a contiguous polyanionic substrate.

We confirmed that our protease-sensitivity assay was reporting structural changes using 2D and 3D classification of AP2 cryo-EM images. In the absence of a DNA oligo, both mu E302K and wild-type AP2 adopted a conformation that matches the crystal structure of the closed complex (PDB 2VGL; *Figure 4C–4E*). In the presence of an anionic membrane mimetic (47 nucleotide DNA),~60% of wild type and more than 90% of AP2 (mu E302K) particles were open (*Figure 4C–4E*, *Figure 4—figure supplement 1*). Similar values are calculated when 3D classification is used (*Figure 4—figure supplement 1C*). On the basis of these results we refer to AP2 (mu E302K) in the presence of anionic polymer as 'open AP2' in our experiments. In addition, this cryo-EM data suggests that the membrane itself affects the conformational equilibrium of AP2 and demonstrates that we can control the conformation of AP2 in vitro using a defined chemical substrate. Using our method to generate open AP2, we confirmed that NECAP bound these complexes in the absence of phosphorylation (*Figure 4F*).

## NECAP$_{Ex}$ recognizes membrane-activated AP2

To identify the domain of NECAP that engages open AP2, we tested NECAP truncations (*Figure 1B*) and found that NECAP$_{Ex}$ was necessary and sufficient for binding (*Figure 5A*). This supports previous data that binding of NECAP$_{PHear}$ to AP2 in rat brain lysate is enhanced by the presence of NECAP$_{Ex}$ (*Ritter et al., 2013*) and explains why NECAP$_{PHear}$ mutants retain this binding (*Beacham et al., 2018*). Because the open state of AP2 precedes phosphorylation (*Conner et al., 2003*; *Hollopeter et al., 2014*), we hypothesize NECAP$_{Ex}$ may form an initial priming interaction with open AP2 prior to phosphorylation.

To visualize NECAP bound to an activated, phosphorylated AP2 complex, we determined a ∼ 3.5 Å structure of phosphorylated AP2 (mu E302K) bound to full-length NECAP2 in the presence of an anionic polymer (DNA) (*Figure 5B*, *Figure 5—figure supplement 1*, *Table 1*). In contrast to other AP2 (mu E302K) complexes incubated with DNA, (*Figure 4B and E*), AP2 was not open, but was in a closed, inactive conformation. As in our previous phosphoAP2-NECAP structure, NECAP$_{PHear}$ was bound to the mu subunit, coordinating pT156. However, in this new structure we observed additional density contacting the beta subunit (*Figure 5B*). Comparing isosurface threshold levels of the unsharpened, refined map clearly shows that this density is contiguous and extends from the C-terminus of NECAP$_{PHear}$ (*Figure 6—figure supplement 1*), suggesting it represents NECAP$_{Ex}$. Because we cannot assign a definitive sequence register to the NECAP$_{Ex}$ density in this structure, we model the most ordered region as a poly-alanine peptide. We believe this structure may represent a post-open, inactive conformation of the AP2 complex, corresponding to phospho-AP2 simultaneously

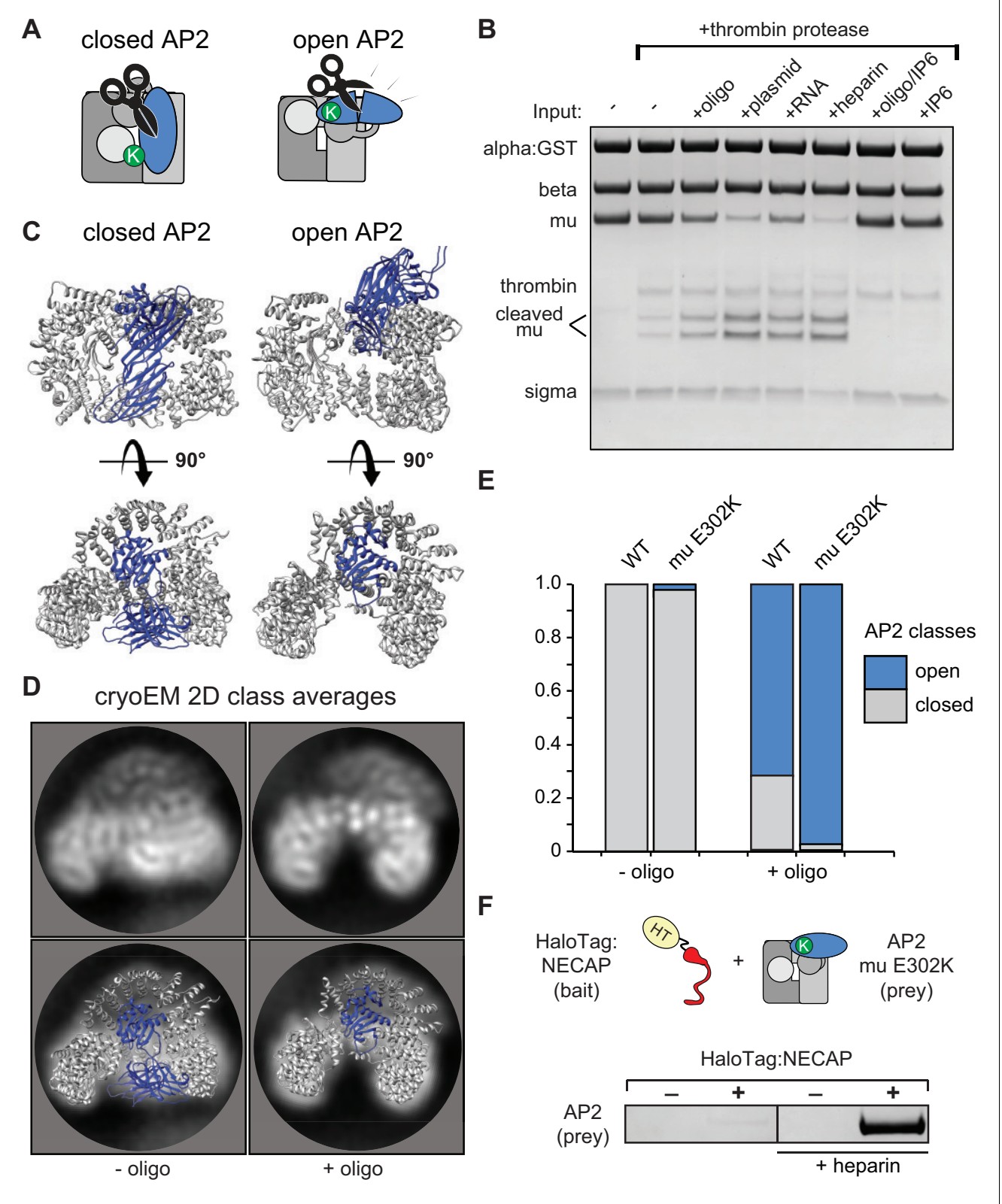

**Figure 4.** Membrane mimetics stimulate opening of AP2. (**A**) Schematic of protease sensitivity assay. Open AP2 exposes thrombin site on mu. K, mu E302K mutation. (**B**) Protease sensitivity of AP2 (mu E302K) in the presence of anionic polymers. Oligo, 60 nucleotide single-stranded DNA; plasmid, double-stranded DNA; RNA, total yeast RNA; IP6: inositol hexakisphosphate. (**C**) AP2 crystal structures (left, PDB 2VGL; right, PDB 2XA7) (**D**) (Top) Representative AP2 cryo-EM 2D class averages in the absence (left) or presence of oligo (47 nucleotide single-stranded DNA, right). (Bottom) Images

*Figure 4 continued on next page*

*Figure 4 continued*

from above overlaid with closed (left) or open (right) AP2 crystal structures. Blue: AP2 mu. The C-terminal domain of mu was omitted from the open crystal structure, as it is disordered in our cryo-EM class averages. (E) Proportion of AP2 particles assigned to either closed (gray) or open (blue) class averages. Data represents ten technical replicates (see definition of technical replicates for this assay in methods). See also *Figure 4—figure supplement 1*. (F) Binding analysis of NECAP to AP2 (mu E302K) in the presence or absence of heparin. HT, HaloTag; K, mu E302K mutation. HaloTag control (-). Representative image of three technical replicates.

DOI: https://doi.org/10.7554/eLife.50003.010

The following figure supplement is available for figure 4:

**Figure supplement 1.** Cryo-EM analysis of AP2 in the presence of an anionic polymer (DNA).

DOI: https://doi.org/10.7554/eLife.50003.011

bound by NECAP and partially engaged with the membrane via the PIP$_2$ pocket on the alpha subunit. We refer to this structure as 'clamped' phosphoAP2-NECAP.

While NECAP$_{Ex}$ appears to make extensive contact with the beta subunit of AP2 in this structure, the differences in AP2 conformation that might account for the appearance of the NECAP$_{Ex}$ binding site are subtle (relative to that lacking DNA, *Figure 1D*, which we refer to as 'unclamped' phosphoAP2-NECAP). The region with the greatest structural changes is the alpha-beta interface, with several helices in the C-terminus of alpha shifting ~1–3 Å (*Figure 5—figure supplement 2*). Additionally, an alpha helix in beta that is part of the Ex domain binding site appears to partially melt when the Ex domain is bound (*Figure 5—figure supplement 2B*). Ordered density for DNA is not

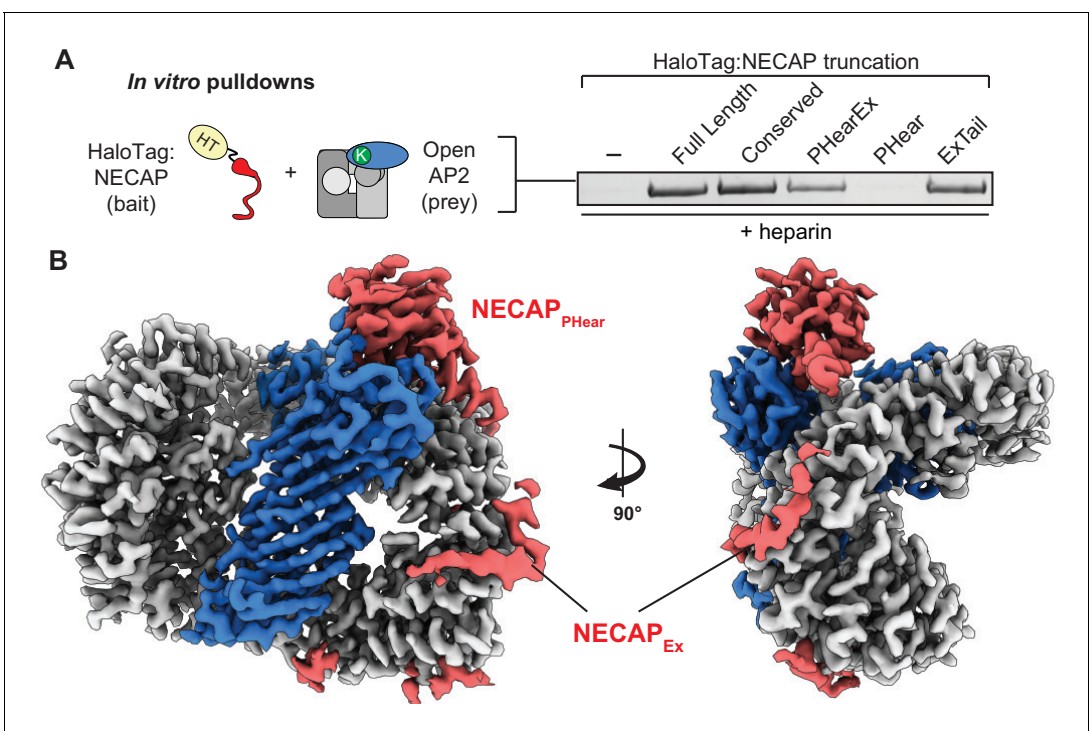

**Figure 5.** NECAP$_{Ex}$ recognizes membrane-activated AP2. (**A**) Binding analysis of NECAP truncations to AP2 (mu E302K) in the presence of heparin. HaloTag control (-). Representative image of two technical replicates. (**B**) Cryo-EM density of the phosphorylated AP2 (mu E302K)-NECAP complex in the presence of oligo (PDB 6OXL, 47 nucleotide single-stranded DNA, five molar excess). See also *Figure 5—figure supplement 1*, *Figure 5—figure supplement 2*.

DOI: https://doi.org/10.7554/eLife.50003.012

The following figure supplements are available for figure 5:

**Figure supplement 1.** Cryo-EM data collection, processing, and model building for phosphoAP2-NECAP-DNA 'clamped' structure.

DOI: https://doi.org/10.7554/eLife.50003.013

**Figure supplement 2.** Structural comparison of 'unclamped' and 'clamped' phosphoAP2-NECAP structures.

DOI: https://doi.org/10.7554/eLife.50003.014

seen at any of the known $PIP_2$ binding sites, consistent with observations that $PIP_2$ binding sites are not ordered pockets but rather a collection of basic residues that protrude into the solvent (*Owen et al., 2004*). Notably, phosphoAP2-NECAP binds to the single-stranded DNA construct used in this experiment with an affinity of ~60 nM (*Figure 4—figure supplement 1*), suggesting that our sample is >95% bound to DNA at the concentrations used to make cryo-EM grids (3 µM protein, 15 µM DNA). Importantly, only the $PIP_2$ binding site on the alpha N-terminus is solvent-exposed in the clamped conformation, so the structural consequences of membrane engagement are fundamentally different in the open (several exposed membrane binding sites) versus the closed (a single membrane binding site) conformations.

After our cryo-EM structure revealed the $NECAP_{Ex}$ binding site, we sought to understand the functional consequences of this interaction. Intriguingly, our genetic screen provided an unexpected insight, as it identified two mutations in the AP2 alpha and beta subunits that escape inactivation by NECAP and are in close proximity to the $NECAP_{Ex}$ binding site (*Figure 2*, *Figure 6A*). Purified AP2 complexes with these mutations have reduced affinity for NECAP in vitro (*Figure 6B*). Additionally, mutation of the most conserved residues of the $NECAP_{Ex}$ domain results in loss of binding to open AP2 (*Figure 6B*) (*Ritter et al., 2013*). All three of these mutations recapitulate the NECAP knockout phenotype in our in vivo fitness assay (*Figure 6C*), imaging assay (*Figure 6D*), and protease sensitivity assay (*Figure 6E*). These data support the conclusion that $NECAP_{Ex}$ specifically recognizes membrane-activated AP2.

## NECAP clamps AP2 in a closed, inactive conformation

Our structural, genetic, and biochemical data suggest a mechanism whereby NECAP clamps AP2 in a closed, inactive conformation via simultaneous engagement of the mu and beta subunits. We tested this hypothesis using our in vitro protease sensitivity assay. We incubated open phosphoAP2 with NECAP and found that including NECAP decreases the protease sensitivity of AP2, consistent with the closed conformation observed in our structure (*Figure 7A*, top). Additionally, by comparing various truncations of NECAP we found that both $NECAP_{PHear}$ and $NECAP_{Ex}$ were required for this activity (*Figure 7A*). When we measured the protease sensitivity of endogenous AP2 in *C. elegans* expressing NECAP truncations, we saw a similar dependence on $NECAP_{PHearEx}$ for full function (*Figure 7A*, bottom). Additionally, we observed that $NECAP_{PHearEx}$ was sufficient to rescue a NECAP deletion in worms using our whole animal fitness assay (*Figure 7B*). Taken together, these data show that $NECAP_{PHearEx}$ locks AP2 into an inactivated conformation.

## Discussion

Our combination of genetic, biochemical, and structural data supports a mechanism by which NECAP inactivates AP2 by initially recognizing open, unphosphorylated complexes and promoting the closed, inactive conformation after phosphorylation (*Figure 7C*). Previous work in vertebrate tissue culture suggests that $NECAP_{Tail}$ mediates binding of AP2 through the alpha appendage (*Ritter et al., 2003*). However, $NECAP_{Tail}$ is poorly conserved in *C. elegans* (*Figure 1B*) and is dispensable for activity in our assays (*Figure 7A,B*) (*Beacham et al., 2018*). This suggests that we have found an additional and perhaps more ancient activity of NECAP, by which the conserved $NECAP_{PHear}$ and $NECAP_{Ex}$ domains cooperate to stabilize the closed conformation of AP2 in a three-step cycle (*Figure 7C*). First, AP2 complexes are recruited to the plasma membrane where $PIP_2$ and other activators induce the opening and stabilization of the complex (*Höning et al., 2005*). Next, $NECAP_{Ex}$ recognizes open AP2 complexes and binds to the beta subunit, placing $NECAP_{PHear}$ close to AP2 at a high local concentration. Finally, after AP2 phosphorylation, $NECAP_{PHear}$ engages the mu subunit of AP2 to clamp the complex in a state resistant to activation (*Figure 7*). Our data confirm this complex is closed despite the otherwise activating signal of membrane mimetics (*Figure 5B*) and is resistant to proteolysis (*Figure 7A*). Importantly, this inactivation cycle has a strong dependence on phosphorylation of AP2 (*Figure 3*), suggesting that inactivation of AP2 is a tightly regulated mechanism.

In general, data from our vertebrate and *C. elegans* systems are consistent, highlighting the evolutionary conservation of the NECAP-AP2 interaction. However, some of the incongruities in our data show the limitations of translating results between model systems. For example, mutation of R112 in $NECAP_{Phear}$ is less disruptive than mutation of mu T156, despite R112 directly coordinating

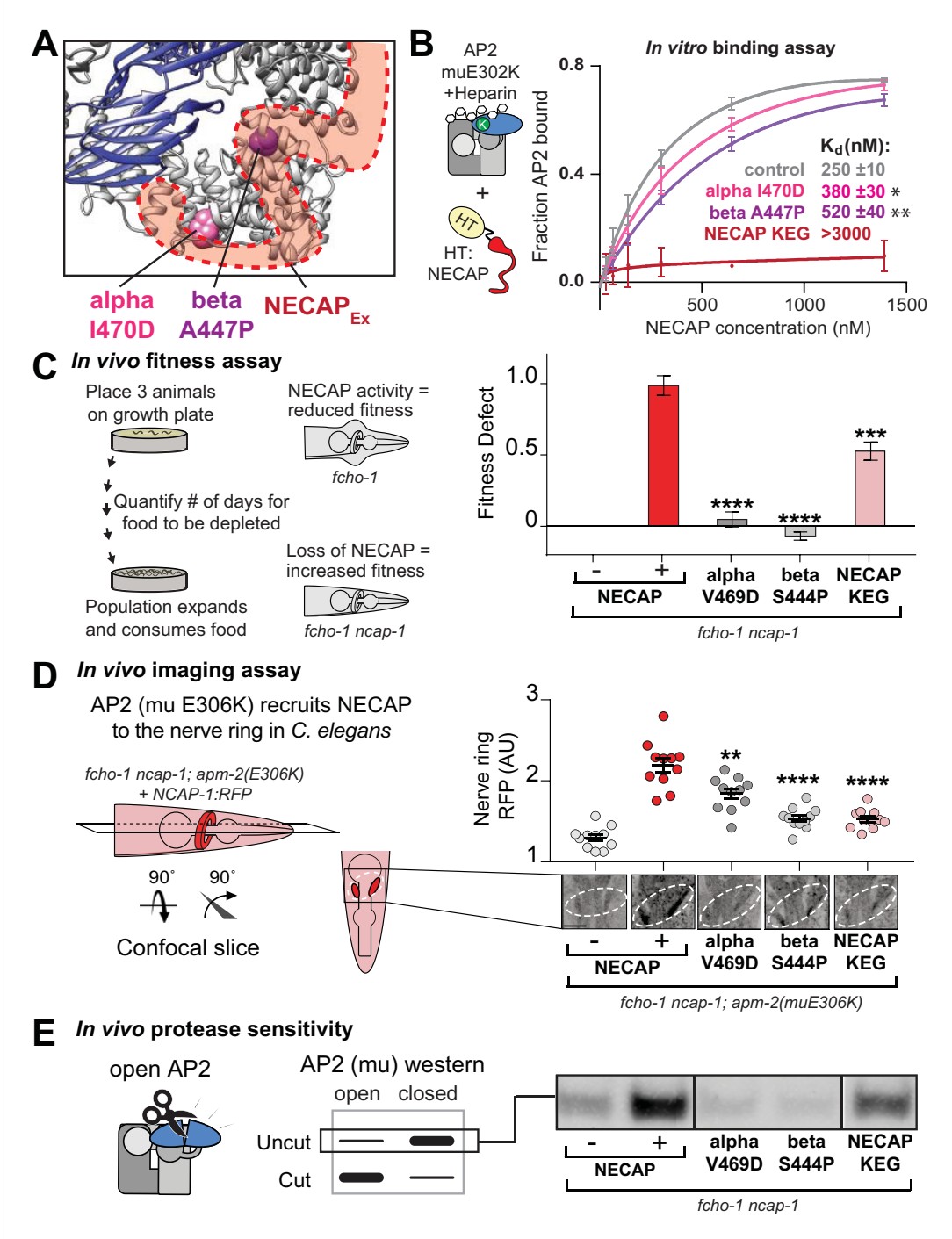

**Figure 6.** The AP2-NECAP$_{Ex}$ interface is required for inactivation. (A) Residues identified in our genetic screen (see *Figure 2*) shown as colored spheres on a ribbon diagram of AP2 with outline of NECAP$_{Ex}$ density (red, low isosurface threshold, see *Figure 6—figure supplement 1*). (B) Binding curves generated from pulldown depletion assays. Error bars represent mean ± SEM from three technical replicates. Inset: Calculated $K_d$ values, variance is SEM. *p<0.05, **p<0.01, relative to control. (C) In the absence of NECAP (–), *fcho-1* mutants take about 4 days to proliferate and consume a bacterial food source (fitness defect = 0). Expression of NECAP (+) increases the number of days to about 8 (fitness defect = 1). Data for interface mutants were normalized to this fitness defect; n = 10 biological replicates. (D) In *fcho-1; apm-2 (E306K)* mutant worms, NECAP is recruited to the nerve ring. Interface mutants disrupt nerve ring recruitment. Normalized RFP intensities plotted above representative confocal nerve ring images of ten biological replicates. (C–D) Error bars indicate mean ± SEM. Significance compared to NECAP (+); Student's t-test performed on raw data (C) or normalized data (D). *p<0.05, **p<0.01, ***p<0.001, ****p<0.0001. (E) In vivo protease sensitivity assay to probe AP2 conformation in genetic backgrounds indicated. In

*Figure 6 continued on next page*

*Figure 6 continued*

the absence of NECAP (–), AP2 is protease sensitive (open). Expression of wild type NECAP (+) results in protease resistant AP2 (closed). All strains lack *fcho-1*. See also *Figure 6—figure supplement 1*.

DOI: https://doi.org/10.7554/eLife.50003.015

The following figure supplement is available for figure 6:

**Figure supplement 1.** The NECAP$_{Ex}$ domain binds along the surface of the beta subunit.
DOI: https://doi.org/10.7554/eLife.50003.016

pT156 in our cryoEM structures (*Figure 2B*). It is possible that R112 is not the sole determinant for NECAP binding, and that other residues also aide in coordinating pT156. Indeed, NECAP R89 lies only 5 Å from mu pT156, suggesting that it may also contribute to coordination of the

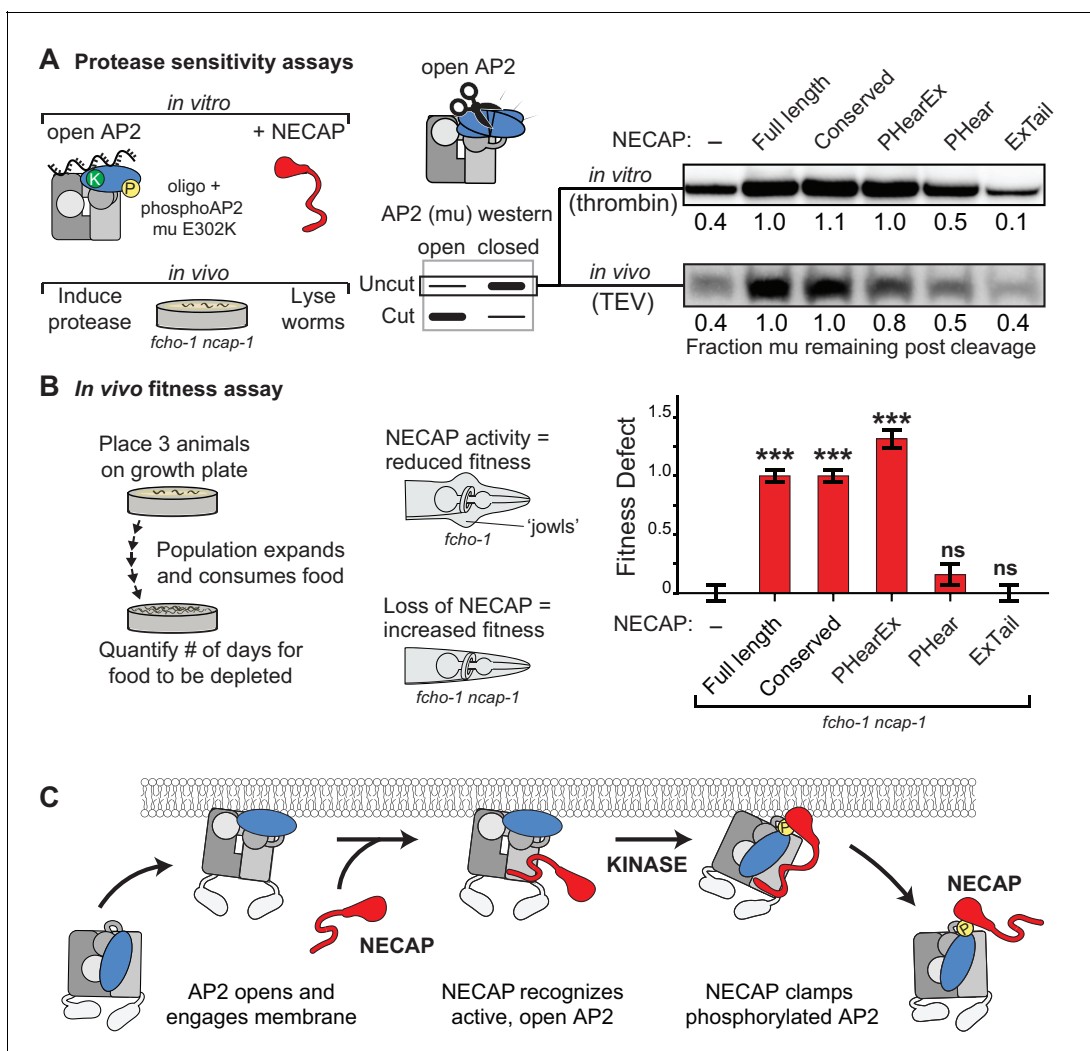

**Figure 7.** NECAP clamps AP2 in a closed, inactive conformation. (**A**) Analysis of NECAP activity on AP2 protease sensitivity. (Left) Schematic of components. Oligo, 60 nucleotide single-stranded DNA. (Center) Samples analyzed by western blot to quantify cleavage of mu subunit. (Right) Western blots cropped to show uncut mu subunit after either addition of protease (in vitro, top) or protease expression via heat shock promoter (in vivo, bottom). Numbers represent normalized intensity of the uncut mu band, relative to the sample with full length NECAP. (**B**) In the absence of NECAP (–), *fcho-1* mutants take about 4 days to proliferate and consume a bacterial food source (fitness defect = 0). Expression of NECAP (Full length) increases the number of days to about 8 (fitness defect = 1). Data for the truncations were normalized to this fitness defect; n = 10 biological replicates. ***p<0.001; ns, not significant; relative to NECAP knockout (-). N = 10 biological replicates. (**C**) Model of AP2 inactivation by NECAP.
DOI: https://doi.org/10.7554/eLife.50003.017

phosphorylation mark. Additionally, we find that the R112 mutation appears less potent in our *C. elegans* assays (*Figure 2D–F*) than in pulldowns using vertebrate proteins (*Figure 2C*) possibly due to evolutionary variance in how NECAP recognizes the phosphate on AP2. Also, mutations that disrupt NECAP binding to phosphoAP2, such as mu T156A, do not fully abolish recruitment of NECAP to AP2 in our nerve ring assay (*Figure 2E*). It is possible that NECAP$_{Phear}$ binding to phosphoAP2 is not a major determinant of NECAP binding in vivo. The residual level of NECAP enrichment could be due to an additional interface between NECAP and AP2, such as NECAP$_{Ex}$ with AP2 beta (*Figure 5*) or NECAP$_{Tail}$ with the AP2 alpha ear (*Ritter et al., 2004*).

Indeed, we have proposed an additional interface in our phosphoAP2-NECAP 'clamped' structure obtained in the presence of nucleic acids, in which we observed density contacting AP2 beta (*Figure 5*). The resolution of our structure (~3.5 Å) is not sufficient to unambiguously assign this density to NECAP, but we believe this density is NECAP$_{Ex}$ for several reasons. First, at low isosurface thresholding, the C-terminus of NECAP$_{PHear}$ appears continuous with the new density (*Figure 6—figure supplement 1*). Second, the surface of AP2 beta in contact with this new density is negatively charged and unlikely to bind polyanions such as DNA. Third, an unbiased genetic screen to identify mutations that reduce activity and binding of NECAP in vivo produced two mutations in AP2 that lie beneath this new density (*Figures 2* and *6A*). However, there still exists the possibility that the density we assign to NECAP$_{Ex}$ actually represents DNA or a portion of NECAP bound to DNA.

If one assumes that our interpretation of the 'clamped' structure is correct, then most of our model (*Figure 7*) is directly supported by cryo-EM structures. An exception is the depiction of NECAP$_{Ex}$ interacting with an open, unphosphorylated AP2 complex. We have not resolved a structure for this complex, and instead propose its existence based on biochemical and in vivo data. AP2 complexes that have been incubated with polyanion membrane mimetics adopt an open conformation according to protease sensitivity and cryo-EM (*Figure 4*). We observe that NECAP$_{Ex}$ binds these complexes in the absence of phosphorylation (*Figure 5A*). We propose that this AP2-NECAP$_{Ex}$ complex occurs early in the AP2 cycle for two main reasons. First, the active state of AP2 appears to induce phosphorylation in vivo, suggesting that AP2 opens on the membrane prior to phosphorylation (*Hollopeter et al., 2014*). Thus, NECAP$_{Ex}$ can engage open AP2 complexes before phosphorylation and subsequent NECAP$_{PHear}$ binding. Second, NECAP is observed early at sites of endocytosis (*Taylor et al., 2011*) and can be incorporated into the coat, apparently without inhibiting pit assembly (*Sochacki et al., 2017*). A priming interaction of NECAP$_{Ex}$ with activated AP2 is consistent with these observations; initially, NECAP is not competent to inactivate AP2, but is poised to do so later in the cycle, after phosphorylation (*Figure 7C*).

While we have uncovered new molecular details of how an endocytic regulator can inactivate the AP2 complex, how this process is controlled both spatially and temporally remains an open question. One possibility is that NECAP action is dictated by the timing of AP2 kinase recruitment or activation (*Conner et al., 2003*; *Jackson et al., 2003*) and that AP2 complexes become inactivated by NECAP immediately upon phosphorylation. Another possibility is that NECAP cannot inactivate stabilized AP2 complexes, but only complexes that are not yet fully initiated, thereby serving as a quality control mechanism (*Aguet et al., 2013*; *Cocucci et al., 2012*; *Ehrlich et al., 2004*; *Puthenveedu and von Zastrow, 2006*). However, NECAP is reported at endocytic structures (*Sochacki et al., 2017*) coincident with the arrival of clathrin (*Taylor et al., 2011*), suggesting that NECAP may act later in the endocytic cycle, after AP2 activation. More speculatively, it is possible that clathrin adaptor inactivation may be involved in vesicle uncoating, as closed AP2 complexes cannot interact with cargo (*Collins et al., 2002*; *Jackson et al., 2010*; *Kelly et al., 2008*) and a clathrin-binding motif on the beta hinge is occluded in this conformation (*Kelly et al., 2014*). The results of this study do not distinguish whether NECAP acts early in endocytosis to promote productive pit formation by limiting aberrant events, or late in the endocytic cycle to uncoat AP2 and allow the complex to initiate new pits. Regardless of when NECAP acts, AP2 must release NECAP and be dephosphorylated to regenerate the steady-state cytosolic pool. It remains to be determined whether NECAP$_{PHear}$ must disengage prior to dephosphorylation, or whether a phosphatase plays a role in NECAP removal from closed complexes.

During review of this manuscript, another study involving AP2, NECAP, and the role of phosphorylation was published (*Wrobel et al., 2019*). This study reports a solution structure of NECAP$_{PHear}$ bound to a fragment of the phosphorylated mu linker (AA 149–163) that agrees with our cryo-EM structures. Their data supports the notion that phosphorylation is a key determinant in the timing

and action of NECAP in the AP2 regulatory cycle, a finding generally consistent with this study. However, Wrobel et al. propose an activating role for AP2 phosphorylation and NECAP, which directly counters our model for NECAP action.

It is also not clear whether NECAP has functions apart from regulation of AP2. For example, the phosphorylated threonine of the AP2 mu subunit is conserved on the mu subunit of AP1 (*Ghosh and Kornfeld, 2003*; *Heldwein et al., 2004*; *Ren et al., 2013*) and NECAP has been proposed to both interact with and control AP1 (*Chamberland et al., 2016*; *Ritter et al., 2004*). This suggests a broader role for NECAP and regulated adaptor inactivation at other membrane compartments.

## Materials and methods

*C. elegans* strains, recombinant proteins, DNA plasmids (including cloning strategies), sequences of synthesized gene fragments (IDT gBlocks), and sequences of DNA oligos are listed in *Supplementary file 1* and *Supplementary file 2*: Key Resources Table.

### Model system

*C. elegans* were maintained at room temperature (22–25℃) using standard procedures (*Brenner, 1974*) on nematode growth medium (NGM) plates seeded with *E. coli* strain OP50.

### Fitness assay

Fitness (starvation) assays were performed as previously described (*Hollopeter et al., 2014*). Briefly, three young adult *C. elegans* hermaphrodites were placed on NGM plates with bacterial food source. The number of days for the population of worms to expand and consume the food was recorded.

### Transgenic strains

Generation of transgenic strains by CRISPR was performed as previously described (*Beacham et al., 2018*) using ribonucleoprotein complexes (*Paix et al., 2015*). Injection mixes contained 19 µM recombinant Cas9 nuclease (purified in-house), 30 µM crispr RNA (crRNA) for desired edit (IDT), 35 µM trans-activating crRNA (trRNA, IDT), 6 µM crRNA targeting the *dpy-10* locus (IDT), and 2.5 µM *dpy-10 (rol)* single stranded oligo repair template (IDT). For small missense mutations, 10 µM single-stranded oligonucleotide (oligo) DNA repair template (IDT) was included. For large insertions, such as fluorophores, repair templates (PCR products with 35 base pair homology arms flanking the Cas9 cleavage site) were included at a final concentration of 1 µM. Gonad arms of young adult hermaphrodite *C. elegans* were injected and *rol* F1 offspring of the injected worms were transferred to a fresh plate, allowed to lay eggs, and genotyped for the desired edit. Non-*rol* F2 offspring were similarly screened to isolate worms with homozygous edits. GUN89 was generated using mos1-mediated single copy insertion as previously described (*Beacham et al., 2018*; *Frøkjær-Jensen et al., 2012*) by injecting targeting vector pEP57 into *C. elegans* strain EG6703.

### Mutagenesis screen

*C. elegans* strains GUN61 or GUN88 (*fcho-1* mutants expressing *RFP:NECAP*) were mutagenized in 0.5 mM N-nitroso-N-ethylurea (ENU, Sigma Aldrich N3385) for 4 hr at 22℃. After washing with M9 buffer, animals were distributed onto growth plates (10 cm NGM plates seeded with concentrated bacterial OP50 culture). Once the worm population had expanded and consumed the food source, ~2 × 2 cm pieces of each plate were transferred to a fresh growth plate. This process was repeated 4–6 times to select for genotypes with greater fitness, which were then visually screened to eliminate NECAP knockouts (RFP negative) and open AP2 mutations (nerve-ring-enriched RFP). Genomic regions corresponding to AP2 subunits and NECAP were amplified by PCR and sequenced in candidate interface mutants. To confirm that the mutations we identified were responsible for the observed phenotypes, we generated them de novo using CRISPR.

## Recombinant protein purification

### AP2 complex purification

AP2 cores were purified as described previously (Hollopeter et al., 2014) with some modifications. Plasmids encoding wild type or mutant AP2 cores were expressed in *E. coli* (BL21 DE3, NEB). To generate phosphorylated AP2 cores, a plasmid encoding the kinase domain of AAK1 was included. *E. coli* cultures (500–6000 mL) expressing the AP2 cores were lysed by sonication in 50 mM Tris pH 8.0, 1000 mM NaCl, 10% glycerol, 10 mM MgCl$_2$, 1 mM CaCl$_2$, 150 ng/μL lysozyme (Sigma), 24 ng/μL DNAse (grade II from bovine pancreas, Roche), 1 mM Phenylmethylsulfonyl fluoride (PMSF), and one cOmplete EDTA-free Protease Inhibitor Cocktail tablet (Roche). Clarified lysate was passed over a column packed with GST resin to bind AP2. Column was washed with 50 mM Tris pH 8.0, 1000 mM NaCl, 10% glycerol, and 1 mM DTT until optical density at 280 nm (OD280) of the flow-through was below 0.05 arbitrary units (AU). Column was then washed in TBS-DTT (20 mM Tris pH 7.6, 150 mM sodium chloride, and 1 mM DTT) prior to elution. Complexes were eluted using glutathione elution buffer (50 mM Tris, 150 mM sodium chloride, 10 mM reduced glutathione, 1 mM DTT, final pH 9.0) or treated with GST-HRV protease (purified in-house) to release AP2 from the GST affinity tag. Eluted complexes were buffer-exchanged with TBS-DTT and concentrated by centrifugal filtration to 1–5 μM before snap-freezing in liquid nitrogen for long-term storage.

### Hexahistidine-tagged (6xHis) protein Purification

Hexahistidine-tagged proteins were purified as described previously (Beacham et al., 2018; Hollopeter et al., 2014) with some modifications. *E. coli* (BL21 DE3, NEB; 500 mL culture) expressing hexahistidine-tagged proteins were lysed by sonication in 50 mM Tris pH 8.0, 500 mM NaCl, 10% glycerol, 10 mM MgCl$_2$, 1 mM CaCl$_2$, 150 ng/μL lysozyme (Sigma), 24 ng/μL DNAse (grade II from bovine pancreas, Roche), 1 mM PMSF, and one cOmplete EDTA-free Protease Inhibitor Cocktail tablet (Roche). Clarified lysate was passed over a column packed with TALON resin (Clontech) to bind the hexahistidine-tagged protein. The column was first washed with 50 mM Tris pH 8.0, 500 mM NaCl, 10% glycerol, and 5 mM BME until the OD280 of the flow-through was below 0.05 AU. Column was then washed with TBS-BME (20 mM Tris pH 7.6, 150 mM sodium chloride, and 5 mM BME), and the bound protein was eluted using TBS-BME supplemented with 150 mM imidazole. Proteins were buffer exchanged with TBS-DTT and concentrated to 100 μM by centrifugal filtration before snap-freezing in liquid nitrogen for long-term storage.

### phosphoAP2-NECAP purification

E.*E. coli* (BL21 DE3, NEB; 500–6000 mL culture) expressing phosphoAP2 core (pGH504/pGH419/pEP82) or phosphoAP2 (mu E302K) core (pGH504/pGB106/pEP82) were prepared as in *AP2 complex purification* above, until after the wash steps. Before elution, 10 mg purified mouse NECAP2:6x-His (pGH503) was flowed over the resin in a total volume of 25 mL TBS-DTT. Column was washed in 50 mM Tris pH 8.0, 500 mM NaCl, 10% glycerol, 5 mM BME. The complex was released from the GST affinity tag by incubation with GST-HRV protease (purified in-house). The phosphoAP2-NECAP complex was further purified using a column packed with TALON resin to remove AP2 that was not bound to 6xHis-tagged NECAP. Column was washed with 50 mM Tris pH 8.0, 500 mM NaCl, 10% glycerol, 5 mM BME. The complex was eluted using 50 mM Tris pH 8.0, 500 mM NaCl, 10% glycerol, and 5 mM BME supplemented with 150 mM imidazole. The sample was concentrated to OD280 of ~4 AU and further purified using gel filtration (Superdex 200 Increase 10/300 GL column, General Electric). The sample was buffer exchanged with 20 mM HEPES pH 8.0, 150 mM KCl, 1 mM DTT during gel filtration. The final complex was concentrated to >10 μM before snap-freezing in liquid nitrogen for long-term storage.

## In vitro pulldowns

Pulldown assays were performed essentially as described (Hollopeter et al., 2014) except the protease cleavage step was 3–6 hr. For the pulldown assay, 80 pmol of purified HaloTag:NECAP bait or HaloTag control, 40 pmol of recombinant AP2 prey, and 10 μL of Magne HaloTag Bead slurry (20%, Promega) were mixed in TBS-DTT (1 mL total volume for each pulldown) and nutated overnight at 4°C. For pulldowns with 'open AP2', 100 μg heparin was also included. After incubation, the beads were washed with TBS-DTT and bound proteins were cleaved from the HaloTag by incubation with

TEV protease (purified in-house, 3–6 hr at 22°C). Eluted proteins were visualized by coomassie-stained sodium dodecyl sulfate polyacrylamide gel electrophoresis (SDS-PAGE). Images in manuscript are cropped to show the largest AP2 band for each prey (alpha:GST, or alpha and beta).

### Pulldown depletion assay

To determine the $K_d$ of NECAP for open AP2, we measured how much AP2 was precipitated by various concentrations of immobilized NECAP. Open AP2 (50 nM AP2 mu E302K with 10 ng/µL heparin) was incubated with HaloTag:NECAP2:6xHis (30–3000 nM) and TALON hexahistidine affinity resin (20 µL) in TBS-T (20 mM Tris pH 7.6, 150 mM NaCl with 0.1% Tween-20, final reaction volume 100 µL). Samples were agitated at 1500 rpm for one hour at 22°C. The resin was allowed to settle and 45 µL of the supernatant was analyzed by coomassie-stained SDS-PAGE.

### In vitro protease assay

This assay was used to measure the conformation of AP2 complexes in vitro. A thrombin protease site (DNA sequence 5'-ctggtgccgcgcggcagc-3') was inserted into the mu subunit after residue S236. This protease site becomes exposed when AP2 adopts the open conformation. To perform the assay, 20 pmol of purified AP2, 3.3 µg of activators (oligo DNA, plasmid DNA, RNA, or heparin), and 50 pmol of HaloTag:NECAP or 50 pmol of HaloTag control were mixed in 30 µL TBS-DTT. Thrombin protease (Sigma T7009, 3 µL) was then added (see amounts and timings below). Reactions were incubated at 22° C for 10 min (0.25 U thrombin, *Figure 4B*) or 30 min (0.5 U thrombin, all others) and terminated by addition of 11 µL 4X bolt LDS Sample Buffer (ThermoFisher) and incubation at 95° C for 5 min. Cleavage of the mu subunit was analyzed by coomassie-stained SDS-PAGE (*Figure 4B*) or by western blot for the phosphorylated mu subunit (*Figure 7A*, top). The primary antibody was rabbit anti-AP2M1 phospho T156 (1:1000, Abcam 109397), and the secondary antibody was goat anti-rabbit Alexa Fluor 647 (1:2000, Life Technologies, A21244).

### In vivo protease assay

In vivo TEV assays were performed as previously described (*Beacham et al., 2018*). *C. elegans* strains used in this assay express heat-shock inducible TEV protease and an AP2 mu transgene containing a TEV protease site that is exposed specifically when AP2 is in an open conformation. The strains also express an RFP:NECAP transgene or RFP alone. 100 L4 stage *C. elegans* hermaphrodites were picked from a growth plate either before or 6 hr after a 1 hr heat shock (34°C). Worms were placed in 1 mL of 20 mM Tris pH 7.6, 150 mM NaCl (TBS) with 0.001% Triton X-100. Animals were pelleted at 1000 g and washed once with TBS containing 0.001% Triton X-100. Samples were pelleted again and the supernatant removed, leaving behind 45 µL total volume. 15 µL 4x bolt LDS Sample Buffer supplemented with fresh dithiothreitol (DTT) was added, and sample was snap frozen in liquid nitrogen. Samples were then lysed in a cup horn sonicator (Branson Ultrasonics Corporation, Danbury, CT; 1 s pulses at 90–95% amplitude for 2–3 min) followed by heating to 70 °C for 10 min. Samples were re-sonicated following the 70 °C denaturation step if any exhibited excessive viscosity. The entire sample was separated by SDS-PAGE and western blot analysis was performed to detect the 3xFLAG-tagged mu subunit. The primary antibody was mouse anti-flag (1:1000, Sigma-Aldrich F3165), and the secondary antibody was goat anti-mouse IRDye 800CW (1:20000, LI-COR, 925–32210).

### In vivo imaging assay

Nerve ring imaging was performed as previously described (*Beacham et al., 2018*) with modifications to data quantification. Live worms were immobilized on slides and imaged on a Zeiss LSM 880 confocal microscope with a 40x water immersion objective. Fluorophores were excited with 488 nm (GFP) and 561 nm (RFP) lasers. Strains for each experiment were imaged in one session with the same laser settings. For each worm, a single confocal slice through the approximate sagittal section of the nerve ring was analyzed in Fiji. The GFP-AP2 signal corresponding to the nerve ring was used to define a region of interest (ROI) for quantification of 'nerve ring RFP'. A second ROI in the anterior of the worm that was outside the nerve ring and pharynx was used for normalization.

## Cryo-EM structure determination

The following conditions were used for all cryo-EM samples. Grids were prepared by glow discharging UltraAuFoil R 1.2/1.3 300 mesh gold grids (Quantifoil GmbH) for 30 s at 20 mAmp. Grids were used within 10 min of charging. 4 µL of sample was applied to grids and plunge frozen in liquid ethane using a Vitrobot Mark IV robot (Thermo Fisher) set to 100% humidity, 4 °C, blot force 20, and blot time 4 s. Samples were imaged using a Talos Arctica TEM (Thermo Fisher) operating at 200 keV in nano probe mode and equipped with a K2 Summit Direct Electron Detector (Thermo Fisher). Parallel illumination of the microscope was performed according to *Herzik et al. (2017)*. Images were collected at 36,000x, yielding a final pixel size of 1.16 Å for counting mode and 0.58 Å for super resolution mode. Dose fractionated movies were collected at a defocus range of −0.6 µm and −2.5 µm and an exposure rate of ~6 e-/pixel/s with 200 ms frames and a total exposure of ~50 e-/Å². New camera gain references were collected before each dataset and the hardware dark reference was updated daily. The microscope was operated using the Leginon software suite (*Suloway et al., 2005*) and data were processed on the fly using the Appion software suite (*Lander et al., 2009*), including motion correction and gain correction using MotionCor2 (*Zheng et al., 2017*), CTF estimation using CTFFIND4 (*Rohou and Grigorieff, 2015*), and particle picking using DogPicker.py (*Voss et al., 2009*). Particles were extracted from dose-weighted, aligned micrographs and analyzed using cryoSPARC (*Punjani et al., 2017*) and Relion-3 (*Nakane et al., 2018*). Unless otherwise noted, resolution values are according to the 0.143 gold standard Fourier shell correlation (GSFSC) method (*Scheres, 2012*).

## phosphoAP2-NECAP cryo-EM structure determination

The phosphorylated AP2 core was purified in complex with full-length mouse NECAP2 using recombinant expression in *E. coli* and used to make cryo-EM grids (see *phosphoAP2-NECAP purification*, above). The final protein buffer was 20 mM HEPES-KCl, pH 8.0, 150 mM KCl, 1 mM DTT. We noticed that particles had a different orientation in the ice based on protein concentration and/or the presence of a detergent, n-Octyl-β-D-Glucopyranoside (β-OG) (*Figure 1—figure supplement 1A*). To increase the angular distribution of the particles in the final dataset, three datasets (5 µM protein, 1 µM protein, and 5 µM protein + 0.05% w/v β-OG) were collected using the same exposure rate, total exposure, frame rate, and magnification, then merged and processed as a single dataset. A total of 1092 movies were collected and 944 remained after removing micrographs with crystalline ice or CTF fits worse than 3.5 Å as judged by the 0.5 criterion in Appion. 890,658 particles were extracted, subjected to multiple rounds of 2D classification and re-extraction in Relion-3, yielding a dataset of 490,560 'clean' particles. These particles were used to generate an *ab initio* 3D model in cryoSPARC, which was used as a search model for 3D classification and 3D auto-refinement in Relion-3. After masking and postprocessing, this yielded a 3.7 Å resolution map. After local CTF refinement and beam-tilt estimation in Relion-3, the resolution improved to 2.9 Å resolution. While the AP2 core was well resolved, the density for NECAP was lower resolution (~4.5 Å local resolution), preventing us from building an accurate model. Signal subtraction (*Bai et al., 2015*) and 3D classification without alignment were used to find a sub-population that refined to 3.2 Å resolution, but with significantly improved density for NECAP. Local resolution estimation was performed using Relion-3, showing a range from 3.1 to 4.2 Å resolution, with the bulk of the model below 3.5 Å. This final map was used for model building, see *Model building and validation*, below.

## phosphoAP2-NECAP (mu E302K) + DNA cryo-EM structure determination

The phosphorylated AP2 core containing a hyper-active mu E302K mutation was purified in complex with full-length mouse NECAP2 using recombinant expression in *E. coli* (see *phosphoAP2-NECAP purification* above). The final protein buffer was 20 mM HEPES-KCl, pH 8.0, 150 mM KCl, 1 mM DTT, 0.05% w/v β-OG. phosphoAP2-NECAP (mu E302K) at 3 µM concentration was mixed with 15 µM of a 60 bp single-stranded DNA oligo (oEP971) and used to make cryo-EM grids. A total of 1497 movies were collected and 1126 remained after removing micrographs with crystalline ice or CTF fits worse than 3.5 Å. 717,231 particles were extracted, subjected to multiple rounds of 2D classification and re-extraction in Relion-3, yielding a dataset of 388,962 'clean' particles. These particles were used to generate an *ab initio* 3D model in cryoSPARC. 324,922 particles were found to go into high-resolution classes using heterogeneous refinement, and a final refinement was performed using the

non-uniform refinement protocol in cryoSPARC v2, yielding a final resolution of 3.5 Å. Map sharpening and local resolution estimation was performed using cryoSPARC v2.

## Model building and validation

For each AP2-NECAP complex, the same general process was followed. The crystal structure of the AP2 complex in the closed conformation (PDB 2VGL) (*Collins et al., 2002*) and the solution structure of mouse NECAP1 (PDB 1TQZ) (*Ritter et al., 2007*) were docked into the cryo-EM map using Chimera (*Pettersen et al., 2004*). First, NECAP was manually rebuilt in Coot to match the cryo-EM density. A round of real space refinement was performed in Phenix using phenix.real_space_refine (*Afonine et al., 2018*). Regions of AP2 absent from the PDB 2VGL model were manually built in Coot (*Emsley and Cowtan, 2004*), including a region of the mu subunit containing phosphorylated T156 (AA154-158). The NECAP model was then improved using RosettaCM and Rosetta *FastRelax* integrated into a cloud-based cryo-EM pipeline (*Cianfrocco et al., 2018*; *Wang et al., 2015*; *Wang et al., 2016*). At this point, a full phosphoAP2-NECAP model was generated and used as a starting model to generate ~1000 models using RosettaCM. The model with the lowest energy score was then used to generate 100 models in Rosetta *FastRelax*. The lowest energy model from this analysis was further refined using phenix.real_space_refine. For the 'unclamped' phosphoAP2-NECAP model, the following side chains were truncated to the Cβ atom (alpha residue 11; beta residue 5; mu residue 253, 261, 379, 380; NECAP residues 62, 102). For the 'clamped' phosphoAP2-NECAP model, the following side chains were truncated to the Cβ atom (alpha residue 11, 123, 217, 341, 381; beta residue 5, 26, 27, 232; mu residue 21, 26, 261, 281, 379, 380; NECAP residues 94, 101, 102, 137). Additionally, a short poly-alanine peptide was modeled into the $NECAP_{Ex}$ density and included during model refinement for the 'clamped' structure.

## 2d and 3d classification of 'open' vs 'closed' AP2 complexes

AP2 and AP2 (mu E302K) were purified using recombinant expression in *E. coli* (see *AP2 complex purification* above). Cryo-EM grids were prepared with 3 µM protein with or without a 5x molar excess of a 60 bp single-stranded DNA oligo (oEP971, same as that used in protease assay). Gel shift assays showed that phosphoAP2 (mu E302K)-NECAP has a Kd of ~60 nM for the DNA oligo used for cryo-EM (=*Figure 4—figure supplement 1D* ). Assuming a single binding site, AP2 is expected to be >98% bound at the concentrations used for our cryo-EM analysis. Four datasets in total were collected: AP2, AP2 + DNA, AP2 (mu E302K), AP2 (mu E302K) + DNA. Particles were extracted and analyzed in Relion-3. For each sample, we first performed several rounds of 2D classification, yielding about ~50% of particles that entered classes with well-resolved secondary structure. After this initial cleaning step, each particle set was randomly split into 10 subsets and subjected to 2D and 3D classification. For 2D classification, subsets were divided into 40 classes. For 3D classification, the initial model was filtered to 30 Å and six classes were used (T = 4, K = 6). Quantification of particles in the 'open' vs. 'closed' conformation is described in the following section.

## Quantification and statistical analysis

### Fitness assays

Measurements of number of days to starve were taken from 10 biological replicates for each strain. Significance was calculated using an unpaired, two tailed T-test with Prism GraphPad software. Data was normalized relative to NECAP (-) and NECAP (+) strains for visualization only.

### Pulldown depletion assays

Three technical replicates were performed for each sample. Alpha:GST band intensity was quantified ($AP2^{MEASURED}$) and these values were plotted against NECAP concentrations on a logarithmic scale. A sigmoidal (4PL) fit was applied to the data using GraphPad Prism 7.04. The asymptote representing the maximum AP2 that would remain in solution at zero NECAP concentration was calculated ($AP2^{MAX}$). The 'fraction AP2 bound' in each sample was then calculated using the formula (1 − [$AP2^{MEASURED}/AP2^{MAX}$]). These normalized values were then plotted against NECAP concentrations on a logarithmic scale and an EC50 ($K_d$) was calculated from a sigmoidal (4PL) fit of the data using GraphPad Prism 7.04. Note that for *Figure 6B*, 'fraction AP2 bound' was plotted against the concentration of NECAP on a linear scale.

## In vivo imaging assays

Strains for each experiment were imaged in one session with the same laser settings. For each worm, a single confocal slice through the approximate sagittal section of the nerve ring was analyzed in Fiji. The GFP-AP2 signal corresponding to the nerve ring was used to define a region of interest (ROI) for quantification of 'nerve ring RFP'. A second ROI in the anterior of the worm that was outside the nerve ring and pharynx was used for normalization. 'Nerve ring RFP' was quantified as the average fluorescent intensity in the nerve ring ROI divided by the average fluorescent intensity in the ROI for normalization. Images were analyzed for ten biological replicates, and significance was calculated using an unpaired, two tailed T-test with Prism GraphPad software.

## 2d and 3d classification of 'open' versus 'closed' AP2 complexes

Four datasets in total were collected: AP2, AP2 + DNA, AP2 (mu E302K), AP2 (mu E302K) + DNA, as described in the previous section. After identifying particles from each dataset that fit into well-defined 2D classes, each particle set was randomly split into 10 subsets (technical replicates) and 2D and 3D classified in Relion-3. For each random subset subjected to 2D classification, classes were divided into 'open' or 'closed' conformations based on projection matching with the 'open' (PDB 2XA7) (*Jackson et al., 2010*) and 'closed' (PDB 2VGL) (*Collins et al., 2002*) crystal structures. Projection matching was performed in SPIDER (*Frank et al., 1996*) with in-house scripts. For 3D classification, classes were determined to be 'open' or 'closed' based on visual inspection and docking of crystal structures using UCSF Chimera. The ten measurements of 'open' vs. 'closed' were used to calculate a standard error of the mean (SEM).

## Data and code availability

The density maps generated during this study are available at the Electron Microscopy Data Bank (EMD-20215, unclamped and EMD-20220, clamped); the atomic structures generated during this study are available at the Protein Data Bank (PDB 6OWO, unclamped and 6OXL, clamped).

## Acknowledgements

We thank E Shen for help with cloning and protein purification; A Joiner, R Feathers, and C Fromme for preliminary negative stain EM; B Brown, C Adler, R Collins, and H Aguilar-Carreno for use of equipment; H Sondermann and W Greentree for technical advice; F Hughson, C Fromme, R Cerione, and C Adler for comments to improve the manuscript; UCSD Cryo-Electron Microscopy Facility, which is supported in part by NIH grants to Dr. Timothy S Baker and a gift from the Agouron Institute to UCSD; UCSD Physics Computing Facility for IT support; Cornell University Biotechnology Resource Center, which is supported by instrument grants NYSTEM CO29155 and NIH S10OD018516.

## Additional information

### Funding

| Funder | Grant reference number | Author |
| --- | --- | --- |
| National Institute of General Medical Sciences | R01 GM127548-01A1 | Gunther Hollopeter |
| Damon Runyon Cancer Research Foundation | DRG-#2285-17 | Richard W Baker |
| National Science Foundation | DGE-1650441 | Gwendolyn M Beacham |

The funders had no role in study design, data collection and interpretation, or the decision to submit the work for publication.

### Author contributions

Edward A Partlow, Conceptualization, Data curation, Formal analysis, Validation, Investigation, Visualization, Methodology, Writing—original draft, Writing—review and editing; Richard W Baker,

Conceptualization, Data curation, Formal analysis, Investigation, Visualization, Methodology, Writing—original draft, Writing—review and editing; Gwendolyn M Beacham, Conceptualization, Formal analysis, Investigation, Visualization, Methodology, Writing—review and editing; Joshua S Chappie, Resources, Supervision, Methodology, Writing—review and editing; Andres E Leschziner, Conceptualization, Resources, Supervision, Project administration, Writing—review and editing; Gunther Hollopeter, Conceptualization, Resources, Supervision, Funding acquisition, Project administration, Writing—review and editing

### Author ORCIDs
Edward A Partlow https://orcid.org/0000-0001-5513-088X
Richard W Baker https://orcid.org/0000-0003-1136-6000
Gwendolyn M Beacham https://orcid.org/0000-0001-7158-6887
Joshua S Chappie https://orcid.org/0000-0002-5733-7275
Gunther Hollopeter https://orcid.org/0000-0002-6409-0530

### Decision letter and Author response
Decision letter https://doi.org/10.7554/eLife.50003.031
Author response https://doi.org/10.7554/eLife.50003.032

## Additional files

### Supplementary files
• Supplementary file 1. Annotated Resources.
DOI: https://doi.org/10.7554/eLife.50003.018

• Supplementary file 2. Key Resources Table.
DOI: https://doi.org/10.7554/eLife.50003.019

• Transparent reporting form
DOI: https://doi.org/10.7554/eLife.50003.020

### Data availability
The density maps generated during this study are available at the Electron Microscopy Data Bank (EMD-20215, unclamped and EMD-20220, clamped). The atomic structures generated during this study are available at the Protein Data Bank (PDB 6OWO, unclamped and 6OXL, clamped).

The following datasets were generated:

| Author(s) | Year | Dataset title | Dataset URL | Database and Identifier |
|---|---|---|---|---|
| Edward A Partlow, Richard W Baker, Gwendolyn M Beacham, Joshua S Chappie, Andres E Leschziner, Gunther Hollopeter | 2019 | Cryo-EM structure of phosphorylated AP-2 core bound to NECAP | https://www.rcsb.org/structure/6OWO | Protein Data Bank, 6OWO |
| Edward A Partlow, Richard W Baker, Gwendolyn M Beacham, Joshua S Chappie, Andres E Leschziner, Gunther Hollopeter | 2019 | Cryo-EM structure of phosphorylated AP-2 (mu E302K) bound to NECAP in the presence of SS DNA | https://www.rcsb.org/structure/6OXL | Protein Data Bank, 6OXL |
| Edward A Partlow, Richard W Baker, Gwendolyn M Beacham, Joshua S Chappie, Andres E Leschziner, Gunther Hollopeter | 2019 | Cryo-EM structure of phosphorylated AP-2 core bound to NECAP | https://www.ebi.ac.uk/pdbe/entry/emdb/EMD-20215 | Electron Microscopy Data Bank, EMD-20215 |
| Edward A Partlow, | 2019 | Cryo-EM structure of | https://www.ebi.ac.uk/ | Electron Microscopy |

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
