## [Decision Letter]

Thank you for submitting your article "A structural mechanism for phosphorylation-dependent inactivation of the AP2 complex" for consideration by *eLife*. Your article has been reviewed by two peer reviewers, and the evaluation has been overseen by Suzanne Pfeffer as the Senior and Reviewing Editor. The reviewers have opted to remain anonymous.

The reviewers have discussed the reviews with one another and the Reviewing Editor has drafted this decision to help you prepare a revised submission.

As summarized by reviewer 1, This submission is a logical and illuminating extension of work from the Hollopeter lab. Using combined structural, biochemical and worm-based approaches, an appealing model for AP2 mu2 adaptin phosphorylation and NECAP binding in returning AP-2 to the closed conformation is proposed. The work documents five substantial advances:

1) Cryo-EM structures of recombinant phosphorylated AP2 core bound to NECAP2 as well as to the non-phosphorylated open form of AP2.

2) The formulation and use of various polymeric negatively charged membrane mimetics to dissect NECAP-AP2 interactions.

3) The identification that the middle "Ex" region of NECAP recognizes the membrane bound open AP2 core without a requirement for phosphorylation.

4) A bivalency model for AP2 closure based upon the data collected.

5) The description that the NECAP clamped AP2 structure functionally deactivates all but the α adaptin PIP2 binding sites.

Overall, the reviewers would be in favor of presentation of the story in *eLife* if you are able to address the concerns of reviewer #2. After carefully evaluating your data he wrote, "I think the authors did an excellent job interpreting their density, but their maps are not so easy to interpret. The challenge in assigning the phospho-Thr156 position, in particular, arises because there is a big gap in the EM map that breaks the connectivity. The authors modeled up to Ser140 for AP-2 mu, but then the density washes out completely, and the authors couldn't resume modeling until ~Gln154. This assignment looks like a good guess because there are three "landmark" residues further down the chain that fit the density well for a ~3A map: Trp161, Arg162, and Try168. In summary, because their map is discontinuous, it is hard to be confident in the register of their atomic model. Nevertheless, their model is parsimonious and a very good guess, based on the fit-to-density and spacing between Trp161, Arg162 and Tyr168. My recommendation is that they tackle this issue head-on and use clearer figures to illustrate how the landmark residues support their decision to resume modeling the chain at Gln154.

My other comments may be harder to address. Given the uncertainty around the modeling of Thr156, their combination of protease accessibility, worm fitness, localization microscopy, and genetic suppressors, altogether, leaves some room for doubt. Finally, the use of 2D cryoEM class averages to support their claim of an induced open state does not convince me. If they have an open state in solution due to interactions with polyanions, they should be able to resolve it in 3D, not just in 2D."

We would welcome a revised manuscript that addresses these issues and/or clarifies the conclusions so that there is no confusion for readers. We hope you will find these comments constructive, and I include the details to help guide your revision process.

*Reviewer #1:*

This submission is a logical and illuminating extension of work from the Hollopeter lab. Using combined structural, biochemical and worm-based approaches, an appealing model for AP2 mu2 adaptin phosphorylation and NECAP binding in returning AP-2 to the closed conformation is proposed. The quality of the data is high. The work is nicely presented and documents of five substantial advances:

1) Cryo-EM structures of recombinant phosphorylated AP2 core bound to NECAP2 as well as to the non-phosphorylated open form of AP2.

2) The formulation and use of various polymeric negatively charged membrane mimetics to dissect NECAP-AP2 interactions.

3) The identification that the middle "Ex" region of NECAP recognizes the membrane bound open AP2 core without a requirement for phosphorylation.

4) A bivalency model for AP2 closure based upon the data collected.

5) The description that the NECAP clamped AP2 structure functionally deactivates all but the α adaptin PIP2 binding sites.

Together, this provides important and novel structural insight into the conformational changes needed for AP2 function.

Because of the rather limited Discussion section, my concerns are more at the level of interpretation and explanation than with the primary data itself. In the Introduction, the authors indicate AP2 inactivation may also regulate abortive events, but do not clearly explain this at the end in light of the structural findings. They do not seriously consider why NECAP is recruited to CCPs quite early, as seen in time-resolved live cell imaging and clearly a steady-state component of bulk CCPs; they vacillate between this being possibly related to abortive disassembly or perhaps indicative of a role later in CME. They do not discuss how multiple site bound NECAP detaches from the locked state to allow dephosphorylation, which must explain the bulk cytosolic population of AP-2 that is closed but dephosphorylated. In other words, they do not explore why NECAP is not bound stoichiometrically to the cytosolic pool of closed AP2.

As far as I can follow, the authors used mouse NECAP2 for their studies. However, there is published data that show NECAP2 is unnecessary for CME, that NECAP1 and -2 have functionally diverged (cannot rescue knockdown of the other paralog), and that NECAP2 instead regulates AP1 during endosomal recycling. Better justification/rationalization of the choice of NECAP2 is required.

There are now three reports of inherited NECAP1 mutations associated with human neurological disease, and also a reported disease-associated NECAP1 mutation in dogs. How these mutations are related to the structural functional findings in this works should be addressed for the general reader.

Reviewer #2:

Partlow et al. report cryoEM structures of the phosphorylated AP-2 complex bound to its regulator, NECAP. AP-2 is thought to cycle between a "closed", inhibited conformation in the cytosol and an "open", membrane-bound state during endocytosis. AP-2 is phosphorylated, but whether phosphorylation promotes AP-2 opening and endocytosis-or AP-2 closing to inhibit endocytosis-remains controversial. A subset of these authors proposed previously that NECAP, a negative regulator of endocytosis, binds to phosphorylated and open-conformation AP-2-and this hypothesis motivated their current cryoEM structural study of NECAP-bound and phosphorylated AP-2. Combined with an impressive set of genetic observations and biochemical binding experiments, this study could clarify NECAP's role in regulating endocytosis through regulatory phosphorylation. However, in its current form the manuscript is confusing and makes what seem like unwarranted leaps in interpretation.

1) At this average resolution, and from the figures presented, I am not yet confident in the model building for AP-2 mu, especially Thr156-Phospho. Would the authors be willing to provide the experimental maps and models for direct inspection? Do the cryoEM maps allow for unambiguous assignment of the amino acid register for AP2 mu subunit residues 154-158? Are the two phospho maps, phosphoAP2-NECAP-DNA 'clamped' versus phosphoAP2-NECAP closed consistent in this region, and do independent modeling efforts cross-validate?

2) Related to point 1, there seem to be mismatches between the structure-based mutagenesis (where R112 plays a major role in coordinating the phosphate), the genetic assays, and the protease sensitivity assay. Specifically, R112 (R109 in worms) was not found as a genetic hit, while A32 (A29 in worms), which is rather far away and not obviously involved in Thr156-phospho binding, was found genetically along with S87 (S84 in worms). S87N and R112E phenocopy in the mouse-based protein pulldown experiment, moreover, but in worms scored for fitness S84N does NOT phenocopy R109E. In the worm nerve ring localization experiment, NECAP R109E and S84N DO phenocopy. Finally, in the protease sensitivity assay, R109E does NOT phenocopy S84N or AP-2 mu T160A. Together with my uncertainty about the register for the structural model, it is unclear to me how to synthesize these observations into an internally consistent model.

3) The use of polyanion membrane mimetics to stabilize conformationally "open" AP-2 lead to further confusion. Specifically, the authors argue that their protease sensitivity assay and 2D class averages reveal how polyanions promote the formation of an open AP-2 complex (although we never see this open state in 3D). They propose that NECAP binds this open state, selectively, in the presence of anions like heparin. Their 3D cryoEM of supposedly activated AP-2 bound to ssDNA and NECAP, however, is of a closed conformation AP-2 that the authors refer to as "post-open." This seems like an unwarranted leap since the structure of the closed-state AP-2 that resolves the NECAPPHEAR domain looks very similar to the structure of the closed-state AP-2 with both NECAPPHEAR and NECAPExTail resolved. Rather, it seems NECAP binds and stabilizes the AP-2 closed state with or without polyanions, and it's hard to know why new, hard-to-interpret cryoEM density resolves in the ssDNA-incubated state. Overall, I am not yet convinced that NECAP binds "open" AP-2 and do not see how the structures reveal NECAPExTail "specificity for open complexes". The cartoon model in Figure 7C illustrates presumed states that are not based on these NECAP-bound structures: the ExTail domain binding to an "open" but not phosphorylated AP-2, and both the PHEAR and ExTail domains bound to an "open" and membrane-associated AP-2.

[Editors' note: further revisions were requested prior to acceptance, as described below.]

Thank you for resubmitting your work entitled "A structural mechanism for phosphorylation-dependent inactivation of the AP2 complex" for further consideration at *eLife*. Your revised article has been favorably evaluated by Suzanne Pfeffer as the Senior and Reviewing Editor, and one reviewer.

The reviewer was convinced by your response to the initial round of comments. He had a few remaining issues however, and we agreed that if you address these directly in a revised discussion (as possible limitations, whether you think they are likely or not), we would be happy to present your story as an *eLife* Research Advance.

The reviewer wrote, "I am convinced that the structural models that include phospho-Thr156 are sound and that the structure of NECAP bound to this moiety is an important advance.

The rest of my comments are, admittedly, nit-picking. I raise them for discussion, not to stand in the way of publishing an interesting study. First, I remain concerned that the functional tests of the structural model were somewhat inconsistent, as I noted previously. Yes, the assays are very different and have different sensitivities and dynamic ranges. But it is still puzzling that the R109E mutant looks more-or-less indistinguishable from WT in the rescue experiments of Figure 3E-F, while mutating Thr156 (160 in the worm) is a clear loss-of-function in 3F, but not in 3E." Please discuss.

"…Also, I remain concerned about the experiments that rely on anionic polymers to mimic PIP2-containing membranes. The authors may be right, DNA and heparin may be transiently inducing structural changes that are akin to membrane-induced opening, but does it follow from this premise that inositol hexakisphosphate should block the effect of these mimetics on AP2? If heparin, RNA and DNA work, why doesn't IP6 also work to open the complex? Because purifying AP2 required high ionic strength solutions (500mM to 1M), but the binding studies done with heparin were done at 150mM NaCl, perhaps non-specific electrostatic "stickiness" may be confounding? A less interesting interpretation of the "post-open" clamped structure is that the new density is non-specifically bound DNA or some combination of a portion of the Ex domain and DNA." Please discuss.

Finally, please include the Molprobity and Clashscore percentiles as well as the absolute scores, as requested previously.

---

## [Author Response]

Overall, the reviewers would be in favor of presentation of the story in eLife if you are able to address the concerns of reviewer #2. After carefully evaluating your data he wrote, "I think the authors did an excellent job interpreting their density, but their maps are not so easy to interpret. The challenge in assigning the phospho-Thr156 position, in particular, arises because there is a big gap in the EM map that breaks the connectivity. The authors modeled up to Ser140 for AP-2 mu, but then the density washes out completely, and the authors couldn't resume modeling until ~Gln154. This assignment looks like a good guess because there are three "landmark" residues further down the chain that fit the density well for a ~3A map: Trp161, Arg162, and Try168. In summary, because their map is discontinuous, it is hard to be confident in the register of their atomic model. Nevertheless, their model is parsimonious and a very good guess, based on the fit-to-density and spacing between Trp161, Arg162 and Tyr168. My recommendation is that they tackle this issue head-on and use clearer figures to illustrate how the landmark residues support their decision to resume modeling the chain at Gln154.

We agree with the reviewer that building the AP2-NECAP interface, and in particular the mu pT156 residue, is vital to our understanding of NECAP function. We acknowledge that how we defined the register of the linker region should be clarified. As suggested by the reviewers, we have added an additional supplementary figure (Figure 1—figure supplement 3) that specifically highlights how we built the mu linker. This shows the docking of the AP2 crystal structure and the 5 residues we built into the new density in our cryoEM map.

In summary, the register of the mu linker is built from an earlier crystal structure of AP2 (PDB 2VGL). This structure is highly congruent with our pAP2-NECAP cryoEM map (Figure 1—figure supplement 2). Additionally, AP2 has been crystallized many times (PDB 2VGL, PDB 2XA7, PDB 2JKR, PDB 4UQI) and all of these models have an identical register in the mu linker region (residues 159-168). Thus, we are confident in the register of mu starting at residue I159. To build our model of the phosphorylated mu linker, we only had to build Q158 and G157 to reach the critical pT156 residue. We hope we have shown this clearly in our new figure (Figure 1—figure supplement 3). We have also added text to the manuscript to clarify these points and direct readers to the new figure.

Results: “The register leading up to this new density is based on a previous crystal structure (PDB 2VGL), and is confirmed by the positions of several landmark residues near T156 that fit the density well. A detailed schematic for building the critical T156 residue in our model is shown in Figure 1—figure supplement 3.”

My other comments may be harder to address. Given the uncertainty around the modeling of Thr156, their combination of protease accessibility, worm fitness, localization microscopy, and genetic suppressors, altogether, leaves some room for doubt.”

We agree that the varying effect of our mutants across our assays is potentially confusing. We have added language in the main text to clarify this point.

Results: “A bona fide interface mutant should reduce the function of NECAP in multiple assays, but not necessarily to the same degree in every assay. This is because each assay exhibits a different linear range, and point mutations may affect *C. elegans* and vertebrate proteins differently.”

Results: “It is worth noting that mutation of R112 did not fully suppress the fitness defect, and thus would likely not have been isolated from our genetic screen. Additionally, chemical mutagenesis favors cytosine to thymine DNA transitions, thus disfavoring charge reversal of R112.”

“Finally, the use of 2D cryoEM class averages to support their claim of an induced open state does not convince me. If they have an open state in solution due to interactions with polyanions, they should be able to resolve it in 3D, not just in 2D."

3D reconstructions of open AP2 complexes in the presence of DNA oligo are shown in Figure 4—figure supplement 1B. Unfortunately, incubation with DNA causes AP2 complexes to have an extreme preferred orientation in the grid, likely due to exposed hydrophobic regions that interact with the air-water interface. This severely limits the resolution of 3D reconstructions. Nonetheless, we can resolve 3D complexes to 7-10 Å resolution, sufficient to distinguish open and closed complexes.

We have added a new panel to Figure 4—figure supplement 1 which quantifies the 3D classification of complexes into open or closed states and corroborates the results obtained using analysis of 2D classes. The following details were added to the manuscript, and clarifying changes were made to “2D and 3D classification of ‘open’ versus ‘closed’ AP2 complexes” in the “Quantification and Statistical Analysis” section.

Results: “Similar values are calculated when 3D classification is used (Figure 4—figure supplement 1C).”

Method details: “After this initial cleaning step, each particle set was randomly split into 10 subsets and subjected to 2D and 3D classification. […] Quantification of particles in the ‘open’ vs. ‘closed’ conformation is described in the following section.”

Figure 4—figure supplement 1 legend: “(C) Quantification of 3D classification. Four datasets were analyzed and the percentage of particles that classified into open or closed 3D classes were quantified and plotted.”

We would welcome a revised manuscript that addresses these issues and/or clarifies the conclusions so that there is no confusion for readers. We hope you will find these comments constructive, and I include the details to help guide your revision process.

Thank you for including the detailed reviewers comments, which we have found helpful in clarifying and improving our manuscript, especially the Discussion. Below is our response to specific reviewer comments.

We have added a paragraph to the Discussion to better explain our model figure, specifically the open AP2-NECAP complex we propose for which we have not resolved a 3D structure. This complex is poorly behaved in solution, and we have been unable to produce concentrated samples for cryo-EM. In addition, micrographs of grids prepared using low concentration samples have not produced classifiable particles. Nonetheless, we believe there is sufficient evidence to propose this structure and place it early in our model.

Discussion: “The model in Figure 7 depicts NECAP first interacting with an open, unphosphorylated AP2 complex through the NECAP_Ex_ domain. […] A priming interaction of NECAP_Ex_ with activated AP2 is consistent with these observations; initially, NECAP is not competent to inactivate AP2, but is poised to do so later in the cycle, after phosphorylation (Figure 7C).”

We have also added several sentences later in the Discussion to address the unanswered questions of when and where NECAP acts, how NECAP detaches from AP2, and how AP2 becomes dephosphorylated.

Discussion: “The results of this study do not distinguish whether NECAP acts early in endocytosis to promote productive pit formation by limiting aberrant events, or late in the endocytic cycle to uncoat AP2 and allow the complex to initiate new pits. […] It remains to be determined whether NECAP_PHear_ must disengage prior to dephosphorylation, or whether a phosphatase plays a role in NECAP removal from closed complexes.”

We clarified our choice of using NECAP2 for our in vitro experiments.

Results: “While there are two paralogues of NECAP in vertebrates, we have previously shown that they function equivalently to bind phosphorylated and open AP2 complexes in vitro and rescue loss of NECAP in *C. elegans* (Beacham et al., 2018). In this work, we use human or mouse NECAP2 for all experiments because of their ease of purification and stability.”

We thank the reviewers for pointing us towards reports of inherited NECAP1 mutations associated with human or canine diseases. While these pathogenic mutations in NECAP are exciting, it is difficult to interpret these mutations in the context of this work. The human variants appear to result in loss of NECAP expression, and the missense mutation observed in Giant Schnauzer dogs is outside the PHearEx region of NECAP that we identified as the minimal functional region in our assays and visualized in our cryo-EM structures.